# The Importance of Boundary Conditions and Failure Criterion in Finite Element Analysis Accuracy—A Comparative Assessment of Periodontal Ligament Biomechanical Behavior

**Radu-Andrei Moga [1,*] , Cristian Doru Olteanu [2,*] and Ada Gabriela Delean [1]**

[1] Department of Cariology, Endodontics and Oral Pathology, School of Dental Medicine, University of Medicine and Pharmacy Iuliu Hatieganu, Str. Motilor 33, 400001 Cluj-Napoca, Romania; ada.delean@umfcluj.ro

[2] Department of Orthodontics, School of Dental Medicine, University of Medicine and Pharmacy Iuliu Hatieganu, Str. Avram Iancu 31, 400083 Cluj-Napoca, Romania

* Correspondence: andrei.moga@umfcluj.ro (R.-A.M.); olteanu.cristian@umfcluj.ro (C.D.O.)





**Featured Application: For a clinician, knowing the amounts of load that can be safely applied during a periodontal breakdown helps in improving the predictability of the orthodontic treatment and avoiding the ischemic and resorptive risks. Thus, knowing that an intact periodontium can bear up to 2.4 N without major ischemic or resorptive risks is of extreme importance. The 4 mm breakdown reference point, after which the applied loads should be lower than 1 N, supplies valuable data for both orthodontics and periodontology. The stress distribution areas displayed for each movement and bone loss level create a general complete image of PDL biomechanical behavior. For a researcher, providing a way to gain the much-needed results for dental studies with an accuracy comparable with those provided by the engineering field is valuable, since FEA is the only available method that allows the individual study of each dental tissue's component, and the current numerical studies have produced debatable and sometimes contradictory results. Thus, by employing the ductile resemblance material type failure criteria (T and VM) and linear elasticity, isotropy, and homogeneity/non-homogeneity as boundary condition assumptions in the study of PDL, the present study obtained results that are in agreement with clinical knowledge. Moreover, the above-mentioned boundary conditions are correct, up to an applied load of 2.4 N (with up to 1 N being acknowledged as mechanically correct).**

**Abstract:** (1) Background: Herein, finite element analysis (FEA) of the periodontal ligament (PDL) was used to assess differences between Tresca (T-non-homogenous) and Von Mises (VM-homogenous) criterion, by simulating a 0–8 mm periodontal breakdown under five orthodontic movements (extrusion, intrusion, rotation, tipping, and translation) and three loads (0.6, 1.2, and 2.4 N). Additionally, we addressed the issues of proper boundary condition selection for more than 1 N loads and correlated the results with the maximum hydrostatic pressure (MHP) and available knowledge, evaluating ischemic and resorptive risks for more than 1 N orthodontic loads. (2) Methods: Eighty-one models of the second lower premolar (nine patients) with intact and 1–8 mm reduced periodontia were created. The assumed boundary conditions were isotropy, homogeneity, and linear elasticity. A total of 486 FEA simulations were performed in Abaqus. (3) Results: Both criteria displayed similar qualitative results, with T being quantitatively 15% higher and better suited. The assumed boundary conditions seem to be correct up to 2.4 N of the applied load. (4) Conclusions: Both criteria displayed constant deformations and displacements manifested in the same areas independently of the load's amount, the only difference being their intensity (doubling—1.2 N; quadrupling—2.4 N). Moreover, 2.4 N seems safe for intact periodontium, while, after a 4 mm loss (seen as the reference point), a load of more than 1 N seems to have significant ischemic and resorptive risks.

**Keywords:** boundary conditions; finite element analysis; periodontal ligament (PDL); orthodontic movements; periodontal breakdown

## 1. Introduction

Finite element analysis (FEA) is the only available method allowing the individual study (both qualitatively and quantitatively) of each dental tissue's component, and it has been introduced in dentistry over the last decade [1]. Being a mathematical algorithm-based method that subdivides a larger structure into smaller and simpler parts called finite elements, it is widely used in the engineering field due to its high accuracy [2]. Nevertheless, its accuracy depends on the accuracy of the input data (i.e., complex geometry, dissimilar material properties, local effects), all acknowledged and known in the engineering field [1].

FEA dental studies [3–19], despite their considerable number, supply various results that differ from one study to another and sometimes contradict clinical data [6–8,20]. If the engineering field benefits from long-established knowledge of materials' physical properties and behavioral types, the dental field does not [1].

FEA begins with the selection of the proper failure criteria (a mathematical algorithm describing the material's behavior as brittle, ductile, or liquid/gas). Ductile materials have a variable ability to deform and recover when subjected to a load, while brittle materials deform little, usually do not recover from deformations, and suffer from cracks and destruction. Dental tissues are considered to be ductile resemblance materials (with a certain brittle flow mode) [1,21–25]. The ductile materials' failure criteria are Tresca (T, maximum shear stress) and Von Mises (VM, maximum overall stress); the brittle criteria are S1 (maximum principal stress) and S3 (minimum principal stress); for liquids, hydrostatic pressure is mandatory [21,22]. In each of these criteria, the variable boundary conditions are the initial parameters that help solve the differential equations and study the biomechanical behavior under specific physical properties (linear elasticity, homogeneity and isotropy vs. non-linear elasticity, non-homogeneity, and anisotropy). The results are also significantly influenced by the analyzed 3D structural model (i.e., anatomical accuracy), obtained either by CBCT's anatomical reconstruction (high accuracy) or artificially created based on anatomical data (low accuracy) [2–4,6–11,14–16,18,19].

The periodontal ligament (PDL) is the most FEA-studied dental tissue component, employing a variety of failure criteria, boundary conditions, and loads, displaying various contradicting results (since none of the above-mentioned issues were thoroughly addressed) [3–19,26].

Anatomically, the PDL's internal micro-architecture consists of collagen fibers displayed as variously oriented dense fiber bundles filling a 0.4–1.5 mm space [22,26], which is of extreme importance for addressing the above issues. The PDL, along with the neurovascular bundle (NVB), are the most sensitive dental tissues to circulatory disturbances due to well-represented vascular support: apical vessels, perforating vessels, and gingival vessels. Outward-facing blood vessels take part in biomechanical suspension and an absorption–dissipation ability, while those that are inward-facing take part in nutritional metabolism [22,26].

Anatomically and biomechanically, dental tissues are non-homogenous, anisotropic, and do not show linear elasticity [23]. However, all dental studies (PDL included) have neglected these issues, assuming linear elasticity, homogeneity, and isotropy (as boundary conditions assumptions), obeying Hook's law due to its easier mathematical equations and a lack of awareness. Nevertheless, from a mechanical point of view (common engineering knowledge), up to a 1 N load and due to small deformations and displacements, all tissues assume linear elasticity and isotropy [13,21–24]. However, for more than 1 N, there are no available data regarding the linear elasticity and isotropy assumptions. Regarding the non-homogeneity/homogeneity issue of dental tissues, the Tresca criterion is suited for ductile non-homogenous cases, while VM is suited for ductile homogenous cases, as correctly reported in the earlier analysis [1,21–25]. There are reports of a quantitative difference between these two criteria, with T being 15–30% higher than VM. The T is more conservative than VM, since it defines a smaller region of elastic behavior in the principal stress state space, being better suited for correct analyses of extremely small complex structures. There are no other FEA studies covering the above issues except our earlier

research [1,21–24]. The type of contact between the model's components (e.g., perfectly bonded interfaces vs. other types) also affects the stress distribution.

The acknowledged dental components' physical properties are as follows: cortical bone 16.7 GPa of compressive modulus and 157 MPa of compressive strength; trabecular/cancellous bone 0.155 GPa of compressive modulus and 6 MPa of compressive strength; enamel 62.2 MPa of compressive stress, 11.5–42.1 MPa of maximum tensile strength, and 53.9–104 MPa of maximum shear stress; dentine 29–73.1 MPa of maximum shear stress; enamel–dentine 53.9–104 MPa of maximum shear stress; PDL maximum tolerable stress 15–26 KPa [27–30].

The orthodontic loads trigger the movements by producing circulatory disturbances in the PDL and NVB-dental pulp [31–35] (i.e., variable changes in the maximum hydrostatic pressure—MHP—of 4.7–16 KPa/26 KPa) [5,16,36–43]. Despite many reports (i.e., numerical and clinical) of both the optimal and maximum amounts of orthodontic load to be safely applied in intact periodontium, it still remains a subject of controversy [3,4,9,10,13,18,25,44–46]. Nevertheless, all these reports agree on the fact that, if the amount of force is too high and/or applied for a longer period, these circulatory disturbances produce ischemia and regressive changes and further resorptive processes [9,10,25,47–49], especially if various bone loss levels are present [49,50]. Regarding the reduced periodontium optimal and maximal amount of orthodontic force, no data were found except in our previous studies [1,21–25]. Moreover, no coherent correlations of MHP and FEA results are available [3–19], except in our previous works [1,21–25].

Most of the current FEA research flow [3–19] has studied intact periodontium models of the upper and lower first molars and upper central incisors, subjected to a limited amount of orthodontic load and one or two movements, providing data limited to this anatomical region. Moreover, there are no FEA studies analyzing all five most common orthodontic movements and their tissular comparative impact. Few numerical studies have investigated the whole premolar area in an intact periodontium [13,14], although various levels of bone loss are relatively common in everyday clinical practice. Moreover, in the orthodontic field, knowing how the bone loss changes the biomechanical stress distribution and how to reduce/keep the amount of applied load in order to avoid ischemia and further tissular loss, is mandatory [1,21–25]; thus, arises the natural scientific need to study the tissular biomechanics of this region during the periodontal breakdown process. Moreover, since the need for a clear image of the biomechanical behavioral changes determined by the bone loss in orthodontic treatment, a gradual horizontal periodontal breakdown process needs to be studied.

Most FEA studies [3–19] usually investigate a model of a single tooth, due to the difficulty of performing numerical simulations on models with multiple teeth [1,21–25]. Field et al. [14] simulated a mandibular arch model with three teeth subjected to orthodontic movements reporting unnatural and clinically incorrect qualitative stress displays, due to the high element size and reduced number of elements-nodes of the analyzed model. To have a correct anatomical model, it needs to be based on CBCT images of in vivo tissues, recorded with at least a 0.075 mm voxel size, but that implies restraining the recorded field. Moreover, a model with a single anatomically accurate tooth (i.e., an extremely small global element size and a high number of elements and nodes) needs extremely high amounts of computation power [1,21–25]. Another issue is related to the fact that there are no algorithms to test the complex tissular biomechanics. In the current dental research flow, there are FEA studies employed for the PDL study S1 [2–4,18], S3 [2,4], VM [2,5,13–19,25], and the hydrostatic pressure [6–12] criterion, with various amounts of loads applied over molar and incisor intact periodontium models, with homogeneity, isotropy, and linear elasticity as boundary conditions, and reporting contradictory results [20,36]. No Tresca studies were available except ours. A small minority of these studies assumed the PDL's non-linearity but used a brittle materials or liquids criterion, with debatable results. Some of these studies reported a 20–50% difference between non-linearity and linearity for less than 1 N loads but employed S1–S3 brittle failure criteria for the PDL (ductile like) [3,4].

Other studies assumed non-linearity/linearity by employing the hydrostatic pressure criterion (with no shear stress) and the Ogden hyper-elastic model for the PDL (despite the internal micro-architectural anatomical reality), with contradictory results from one study to another [3,4,6–10,14], and firmly reporting that only the hydrostatic pressure can be used in the FEA studies [5]. All these studies proved a lack of awareness about the requirements of the FEA method applied in dentistry, not addressing the above-mentioned issues.

Only by correctly identifying and employing the boundary conditions and the proper failure criterion, can a FEA dental study become as reliable and correct as those in the engineering field [21,22]. In previous comparative studies, our team proved that dental tissues have ductile-like resemblances and that only VM and T criteria supply accurate results [1,21–25]. Only one other older FEA-based study approached the issue of the proper failure criteria, but for brittle-like root canal filling, correlating the failure criterion with the analyzed material type (which is mandatory in the engineering field) [2]. No other FEA studies approached these extremely critical issues for FEA accuracy.

Herein, the aims were (a) to qualitatively and quantitatively assess the differences between T (non-homogenous) and VM (homogenous) failure criterion by simulating 0–8 mm periodontal breakdown under five orthodontic movements (extrusion, intrusion, rotation, tipping and translation) and three loads (0.6, 1.2 and 2.4 N) in PDL; (b) to verify the use of linear elasticity, isotropy, and homogeneity/non-homogeneity as boundary conditions assumptions for more than 1 N loads for PDL; and (c) to correlate the quantitative results with MHP and available clinical knowledge, evaluating the ischemic and resorptive risks for more than 1 N orthodontic loads in PDL.

## 2. Materials and Methods

The present numerical study is part of a larger step-by-step developed project [1,21–25] (clinical protocol 158/2 April 2018) assessing the proper finite elements method and boundary conditions to investigate the biomechanical behavior of dental tissues during orthodontic movements and various bone loss levels.

Here, we ran 486 numerical simulations for FEA, analyzing 81 lower premolar models from nine patients (mean age 29.81 ± 1.45 years, 4 males, 5 females, oral informed consent). Our sample size was nine, more than the above-mentioned FEA studies that used a sample size of one (since numerical studies require only a small sample size).

The including criteria were a complete mandibular dental arch in the region of interest, with no malposition and intact teeth (no endodontic treatment, no filling or crown) in the analyzed region, no advanced bone loss, non-inflamed periodontium, orthodontic treatment indication, and proper oral hygiene. The non-suitable patients were rated as such due to their incomplete mandibular arch, malposition, or non-intact teeth, in the region of interest. From the patients that were included in the study, other exclusion criteria were considered to be less common root geometry (e.g., non-fused double rooted, angulated root, root extreme curvature etc.), an abnormal shape of the crown, deciduous teeth, abnormal root surface defects (e.g., external root resorption) or bone shape (various types of bone defects radiological visible), an abnormal pulp chamber (internal resorption, radiologically identified), bone loss of more than 2–3 mm, any signs of inflamed periodontium or bad oral hygiene after the acceptance in the study.

The region of interest was the mandibular lateral region with the two molars and premolars, which was analyzed by means of CBCT (cone beam computed tomography, ProMax 3DS, Planmeca, Helsinki, Finland, 0.075 mm voxel size). The reconstruction software was Amira 5.4.0 (Visage Imaging Inc. 300 Brickstone Square, Suite 201 Andover, MA, USA). The reconstruction was manually performed, since the automated process did not identify and separate the various shapes of grey anatomical tissues on DICOM slices.

The analyzed model included the second lower premolar, with the surrounding bone, PDL, and NVB, while the alveolar sockets of the other three teeth were filled with cortical and trabecular bone. The manual reconstruction process identified and reconstructed all dental tissue components, including enamel, dentine, dental pulp, NVB, cortical bone, and

trabecular bone. The cementum component failed to be clearly separated from the dentine component; thus, it was reconstructed as dentine, due to their similar physical properties (Table 1). All components were assembled into a single 3D mesh model (Figure 1). The PDL has a variable thickness of 0.15–0.225 mm, with NVB included in the apical third. On the vestibular side of the crown (enamel surface), the base of a stainless-steel bracket was reconstructed. For each of the nine patients, a mesh model with limited various bone loss levels (limited to the cervical third) was obtained. The nine second premolar models included seven that were single-rooted and three that were fused and double-rooted. The missing bone and PDL were manually reconstructed to obtain a mesh model with intact periodontium. The mesh models had 5.06–6.05 million C3D4 tetrahedral elements, 0.97–1.07 million nodes, and a global element size of 0.08–0.116 mm (i.e., a high level of anatomical accuracy, when compared with the above-mentioned FEA studies).

The manual reconstruction displayed a limited number of surface irregularities in all models, located in non-essential areas (i.e., quasi-continuous stress areas) (Figure 2). All internal quality control checking algorithms were successfully passed, with no element errors, and with only a limited number of element warnings (e.g., the maximum number of elements warnings was 264 [0.0043%] for a total number of 6.05 million elements). The topographical mesh distribution for elements warnings was 201 (0.0039%) of 5,117,355 for the bone, 63 (0.00677%) of 930,023 for the tooth-bracket-PDL, 39 (0.00586%) of 665,501 for the tooth bracket, 26 (0.00459185%) of 566,221 for the dentine, and 17 (0.0141469%) of 120,168 for the enamel–bracket elements. The internal controls and safety checks of both software models (i.e., the image reconstruction and the finite element analysis) do not allow the next phase of the process if errors or too many anomalies are present, an aspect that was not seen in our study.

Each of the nine models was subjected to a gradual horizontal periodontal breakdown of 1 mm, ranging from 0 to 8 mm bone loss, thus obtaining a total number of eighty-one models with intact and reduced periodontia.

The FEA simulations (a total of 486) were performed in Abaqus 6.13-1 software (Dassault Systèmes Simulia Corp., Stationsplein 8-K, 6221 BT Maastricht, The Netherlands). Five orthodontic movements were simulated over dental tissue (extrusion, intrusion, rotation, tipping, and translation) under three loads of 0.6 N (approx. 60 gf), 1.2 N (approx. 120 gf), and 2.4 N (approx. 240 gf). The boundary condition assumptions were isotropy, homogeneity, and linear elasticity (as in all above-mentioned FEA studies), with perfectly bonded interfaces and the base of the models enclosed (Figure 2). Fixed boundary conditions (i.e., encastre) were imposed at the bottom of the models (i.e., the translations of the FE nodes were restricted; thus, the rotations were also zero). To our understanding, these conditions are representative of and relevant to the studied issues. The failure criterion used were the non-homogeneous ductile materials' Tresca and the homogenous ductile materials Von Mises criteria. The load manager conditions selection was as follows: step procedure—static, general; load type—pressure; load status—created in step, distribution—uniform; magnitude—depending on the surface area; and amplitude—ramp. The Abaqus boundary condition manager selection was as follows: step procedure—static, general; boundary condition type—symmetry/antisymmetric/encastre; boundary condition status—created in step. Although the numerical domain was large (around 6 million FE), the analysis time was reasonable, due to the following considerations: linear elastic analysis (no change in structural stiffness), a small amount of force per each load increment (thus, the stress concentrations are avoided), stable mesh with a negligible number of poor shaped FE, and perfect bond between the biological parts.

The load selection of 0.6 N was motivated by the fact that it was lower than 1 N (mechanically important), and closer to the loads selected for other simulations in our research project. 1.2 N was selected, since it was a little bit higher than 1 N and was double the first one. 2.4 N was selected, being the double of the second one, and closer to the other above-mentioned FEA studies, to enable correlations.

The simulations displayed qualitative (color-coded projections of the maximum amount of stress—red/orange = high, yellow–green = moderate, blue = low) and quantitative (in KPa) results for PDL and correlated them with the 16 KPa of MHP to evaluate the ischemic and resorptive risks during the periodontal breakdown process.

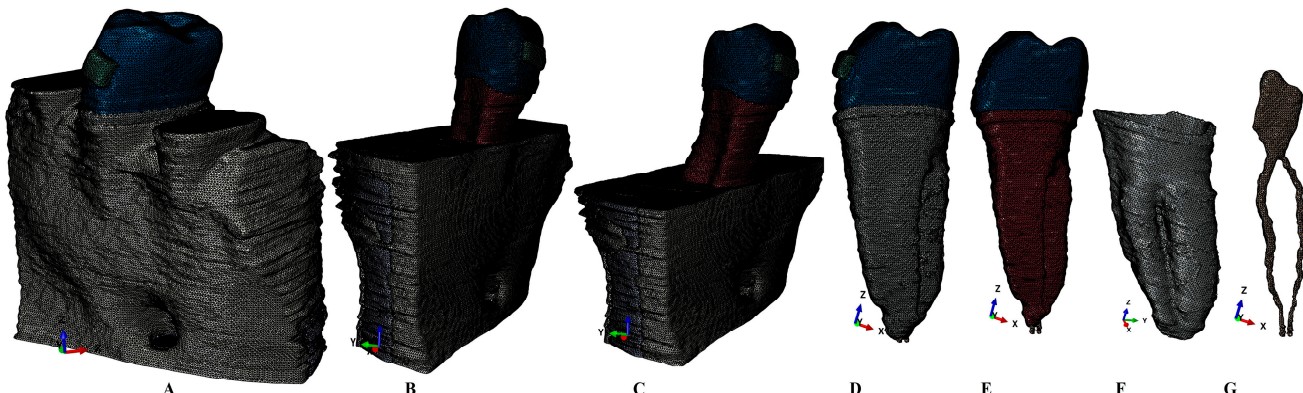

**Figure 1.** Mesh model: (**A**)—2nd lower right premolar model with 0 mm bone loss, (**B**)—2nd lower right premolar model with 4 mm bone loss, (**C**)—2nd lower right premolar model with 8 mm bone loss, (**D**)—2nd lower right premolar model with PDL, dental pulp, NVB, and bracket, (**E**)—2nd lower right premolar model with dental pulp and NVB, (**F**)—intact PDL, (**G**)—dental pulp and NVB.

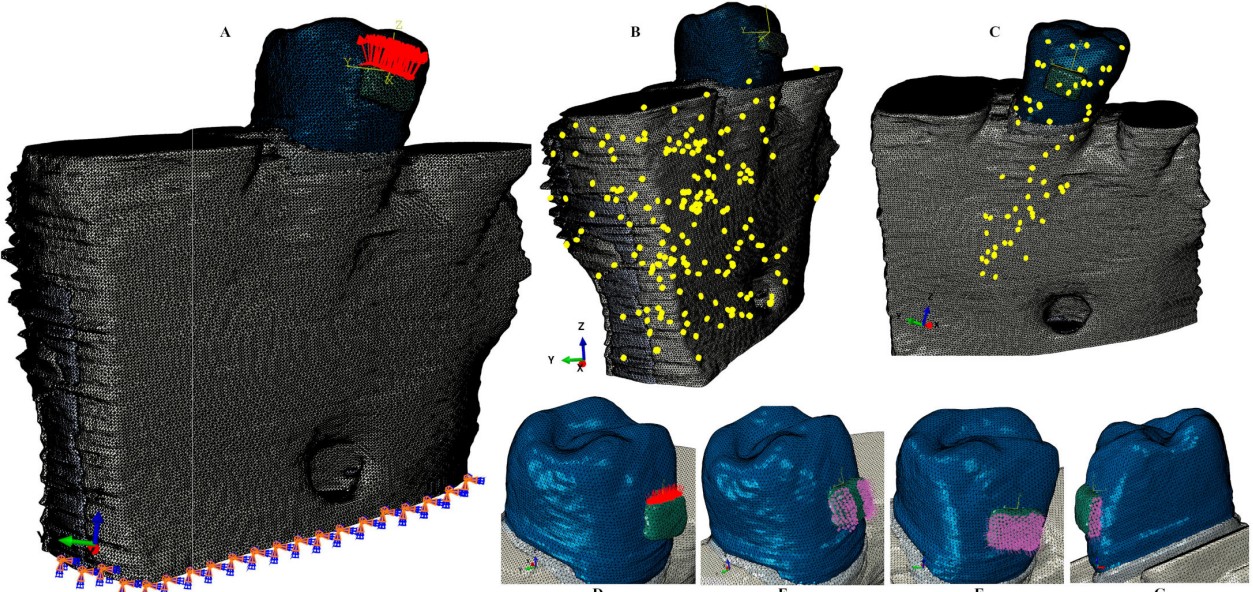

**Figure 2.** Mesh model: (**A**) extrusion vectors and encastered base, (**D**)—intrusion, (**E**)—rotation, (**F**)—tipping, (**G**)—translation; element warnings in one of nine intact periodontium models (**B**)—bone, (**C**)—tooth and PDL.

**Table 1.** Elastic properties of materials.

| Material | Young's Modulus, E (GPa) | Poisson Ratio, $v$ | Refs. |
|---|---|---|---|
| Enamel | 80 | 0.33 | [1,21–25] |
| Dentin/cementum | 18.6 | 0.31 | [1,21–25] |
| Pulp | 0.0021 | 0.45 | [1,21–25] |
| PDL | 0.0667 | 0.49 | [1,21–25] |
| Cortical bone | 14.5 | 0.323 | [1,21–25] |
| Trabecular bone | 1.37 | 0.3 | [1,21–25] |
| Stainless steel | 190 | 0.265 | [1,21–25] |

## 3. Results

The present FEA totaled 486 simulations, displaying qualitative and quantitative results for PDL for both criteria.

The qualitative results displayed similar stress distribution for both criterions. There were no differences in the stress display among the three loads.

Quantitatively, the Tresca displayed amounts of stress that were 15% higher than Von Mises. For both criteria, the maximum shear and overall stresses doubled to 1.2 N and quadrupled to 2.4 N, when compared with 0.6 N. The highest amounts of stress were displayed by the rotation and translation, followed by the tipping, extrusion, and intrusion. The cervical third stress always displayed the highest amounts of stress, independently of the movements, bone loss levels, loads, and criteria. The amount of stress increase is directly correlated with the periodontal breakdown. The rotation and translation, followed by tipping, are the most stressful movements prone to higher ischemic and resorptive risks (exceeding the 16 KPa of MHP) than the extrusion and intrusion, especially after 4 mm of bone loss. The 0.6–1.2 N seems perfectly safe to be applied in both intact and reduced periodontium. However, after 1.2 N, the cervical third stress progressively increased, which correlated with load and bone loss levels, extending to the middle and apical thirds, thus increasing the ischemic and resorptive risks (especially for the rotation, translation, and tipping). However, it must be kept in mind that these are pure orthodontic movements, while clinically it is usually an association and combination that could lead to lower amounts of stress than here. These aspects were considered when the ischemic and resorptive risks were evaluated.

The assumed boundary conditions (i.e., isotropy, homogeneity, linear elasticity, and perfectly bonded interfaces) that were acknowledged as being correctly applied from the physical mechanical point of view for up to 1 N of load, seemed to be correct up to 2.4 N of load, for both criteria. Since the Tresca algorithm was designed for non-homogenous materials, it seems better suited for the study of PDL, falling within the range specified in the literature of 15–30% (when compared with the VM). The Von Mises criteria seemed also suited for dental tissue FEA studies, despite its homogenous material mathematical algorithm design, if all the above-mentioned issues are acknowledged.

Regarding extrusion (Figure 3, Table 2), qualitatively, the stress display areas were similar for both criteria, bone levels and loads, with an extension in the entire PDL (from apical to cervical thirds). Quantitatively, for both criteria, in an intact periodontium experiencing up to 2.4 N, the orthodontic loads were tolerable for the PDL, with the highest amount of stress concentrated in the cervical third. In a reduced periodontium, the stress increase was correlated with bone loss. 0.6 N was perfectly tolerable up to 8 mm bone loss. For 1.2 N, the cervical third stress progressively increased after 4 mm of loss, up to 40–45 KPa (2.8 times higher than MHP). However, both the apical and middle PDL thirds (with richer vascularity than the cervical third) displayed stresses lower than MHP. 2.4 N displayed overruns of the MHP in the entire PDL after 4 mm of loss.

**Table 2.** T and VM maximum stress average values (Kpa) produced by extrusion in PDL.

| Resorption (mm) | | | 0 | 1 | 2 | 3 | 4 | 5 | 6 | 7 | 8 |
|---|---|---|---|---|---|---|---|---|---|---|---|
| Extrusion | T | a | 3.00 | 3.49 | 3.97 | 4.46 | 4.96 | 5.77 | 6.59 | 7.40 | 8.25 |
| 0.6 N/60 gf | | m | 3.00 | 3.80 | 4.59 | 5.39 | 6.18 | 6.70 | 7.22 | 7.73 | 8.25 |
| | | c | 6.70 | 8.41 | 10.12 | 11.82 | 13.52 | 15.61 | **18.60** | 20.52 | **22.50** |
| | VM | a | 2.60 | 3.04 | 3.47 | 3.91 | 4.35 | 5.06 | 5.77 | 6.49 | 7.20 |
| | | m | 2.60 | 3.31 | 4.01 | 4.72 | 5.42 | 5.87 | 6.32 | 6.77 | 7.20 |
| | | c | 5.82 | 7.33 | 8.84 | 10.35 | 11.85 | 14.10 | **16.34** | 18.59 | **19.61** |
| 1.2 N/120 gf | T | a | 6.00 | 6.98 | 7.95 | 8.93 | 9.92 | 11.54 | 13.18 | 14.81 | 16.50 |
| | | m | 5.99 | 7.59 | 9.18 | 10.77 | 12.37 | 13.40 | 14.43 | 15.47 | 16.50 |
| | | c | 13.41 | 16.82 | **20.23** | **23.64** | **27.05** | **31.22** | **37.20** | **41.04** | **45.00** |

**Table 2.** *Cont.*

| Resorption (mm) | | | 0 | 1 | 2 | 3 | 4 | 5 | 6 | 7 | 8 |
|---|---|---|---|---|---|---|---|---|---|---|---|---|
| | VM | a | 5.21 | 6.08 | 6.95 | 7.82 | 8.70 | 10.12 | 11.54 | 12.97 | 14.39 |
| | | m | 5.21 | 6.61 | 8.02 | 9.43 | 10.84 | 11.74 | 12.64 | 13.55 | 14.39 |
| | | c | 11.64 | 14.66 | **17.68** | 20.69 | **23.71** | 28.20 | **32.69** | 37.18 | **39.22** |
| 2.4 N/240 gf | T | a | 11.99 | 13.95 | 15.90 | **17.85** | **19.84** | 23.09 | **26.35** | 29.62 | **33.00** |
| | | m | 11.99 | 15.18 | **18.36** | 21.54 | 24.73 | 26.79 | 28.86 | 30.93 | 33.00 |
| | | c | **26.81** | **33.65** | 40.46 | 47.28 | 54.10 | 62.45 | 74.40 | 82.08 | 90.00 |
| | VM | a | 10.42 | 12.15 | 13.90 | 15.63 | **17.40** | 20.23 | 23.09 | 25.94 | 28.78 |
| | | m | 10.41 | 13.22 | 16.05 | **18.86** | 21.69 | 23.48 | 25.29 | 27.10 | 28.79 |
| | | c | **23.29** | 29.32 | 35.35 | 41.39 | 47.41 | 56.40 | 65.38 | 74.35 | 78.43 |
| Extrusion | T/VM % | a | 1.15 | 1.15 | 1.14 | 1.14 | 1.14 | 1.14 | 1.14 | 1.14 | 1.15 |
| | | m | 1.15 | 1.15 | 1.14 | 1.14 | 1.14 | 1.14 | 1.14 | 1.14 | 1.15 |
| | | c | 1.15 | 1.15 | 1.14 | 1.14 | 1.14 | 1.11 | 1.14 | 1.10 | 1.15 |

T—Tresca, VM—Von Mises, a—apical third, m—middle third, c—cervical third, T/VM%—% increase.

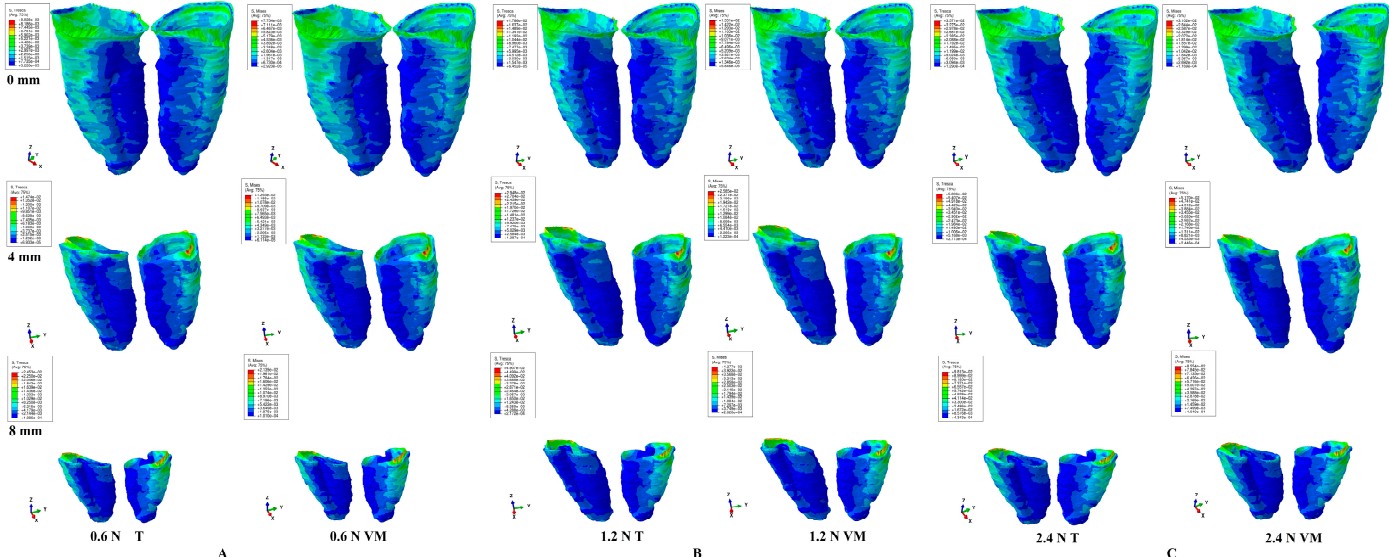

**Figure 3.** Extrusion—Tresca and VM qualitative and quantitative stress display in PDL for 0.6 N (**A**), 1.2 N (**B**), and 2.4 N (**C**), and 0-, 4-, and 8-mm periodontal breakdowns.

For intrusion (Figure 4, Table 3), quantitatively, in both intact and reduced periodontia, the intrusion displayed similar biomechanical behavior to the extrusion. However, qualitatively, the red–orange high stress areas displayed by the cervical third were more extended than for extrusion, for the entire periodontal breakdown.

**Table 3.** T and VM maximum stress average values (Kpa) produced by intrusion in PDL.

| Resorption (mm) | | | 0 | 1 | 2 | 3 | 4 | 5 | 6 | 7 | 8 |
|---|---|---|---|---|---|---|---|---|---|---|---|---|
| Intrusion 0.6 N/60 gf | T | a | 3.00 | 3.49 | 3.98 | 4.46 | 4.96 | 5.78 | 6.62 | 7.43 | 8.25 |
| | | m | 3.00 | 5.21 | 5.51 | 6.31 | 7.41 | 7.62 | 7.83 | 8.04 | 8.25 |
| | | c | 5.22 | 6.68 | 8.15 | 9.61 | 11.07 | 12.91 | 14.75 | 16.58 | **18.43** |
| | VM | a | 2.60 | 3.03 | 3.45 | 3.88 | 4.35 | 5.06 | 5.80 | 6.51 | 7.20 |
| | | m | 2.60 | 4.52 | 4.78 | 5.47 | 6.49 | 6.68 | 6.87 | 7.05 | 7.19 |
| | | c | 4.54 | 5.81 | 7.08 | 8.35 | 9.71 | 11.32 | 12.93 | 14.54 | 16.06 |
| 1.2 N/120 gf | T | a | 5.99 | 6.98 | 7.95 | 8.93 | 9.92 | 11.55 | 13.23 | 14.85 | 16.50 |
| | | m | 5.99 | 10.41 | 11.01 | 12.62 | 14.81 | 15.24 | 15.66 | 16.08 | 16.50 |
| | | c | 10.44 | 13.37 | 16.30 | **19.22** | **22.15** | 25.82 | **29.49** | 33.17 | **36.86** |

**Table 3.** *Cont.*

| Resorption (mm) | | | 0 | 1 | 2 | 3 | 4 | 5 | 6 | 7 | 8 |
|---|---|---|---|---|---|---|---|---|---|---|---|
| | VM | a | 5.21 | 6.06 | 6.91 | 7.75 | 8.70 | 10.13 | 11.60 | 13.02 | 14.39 |
| | | m | 5.20 | 9.03 | 9.55 | 10.94 | 12.99 | 13.36 | 13.73 | 14.10 | 14.39 |
| | | c | 9.07 | 11.62 | 14.16 | 16.70 | **19.42** | **22.63** | **25.86** | **29.08** | **32.13** |
| 2.4 N/240 gf | T | a | 11.99 | 13.95 | 15.90 | **17.85** | **19.84** | **23.10** | **26.46** | **29.70** | **33.00** |
| | | m | 11.99 | **20.83** | **22.02** | **25.23** | **29.62** | **30.48** | **31.32** | **32.16** | **33.00** |
| | | c | **20.88** | **26.73** | **32.59** | **38.43** | **44.29** | **51.63** | **58.99** | **66.33** | **73.71** |
| | VM | a | 10.42 | 12.12 | 13.81 | 15.51 | **17.40** | **20.26** | **23.20** | **26.04** | **28.78** |
| | | m | 10.40 | **18.07** | **19.10** | **21.89** | **25.97** | **26.72** | **27.46** | **28.20** | **28.77** |
| | | c | **18.15** | **23.23** | **28.33** | **33.40** | **38.83** | **45.27** | **51.72** | **58.15** | **64.25** |
| Intrusion | T/VM % | a | 1.15 | 1.15 | 1.15 | 1.15 | 1.14 | 1.14 | 1.14 | 1.14 | 1.15 |
| | | m | 1.15 | 1.15 | 1.15 | 1.15 | 1.14 | 1.14 | 1.14 | 1.14 | 1.15 |
| | | c | 1.15 | 1.15 | 1.15 | 1.15 | 1.14 | 1.14 | 1.14 | 1.14 | 1.15 |

T—Tresca, VM—Von Mises, a—apical third, m—middle third, c—cervical third, T/VM%—% increase.

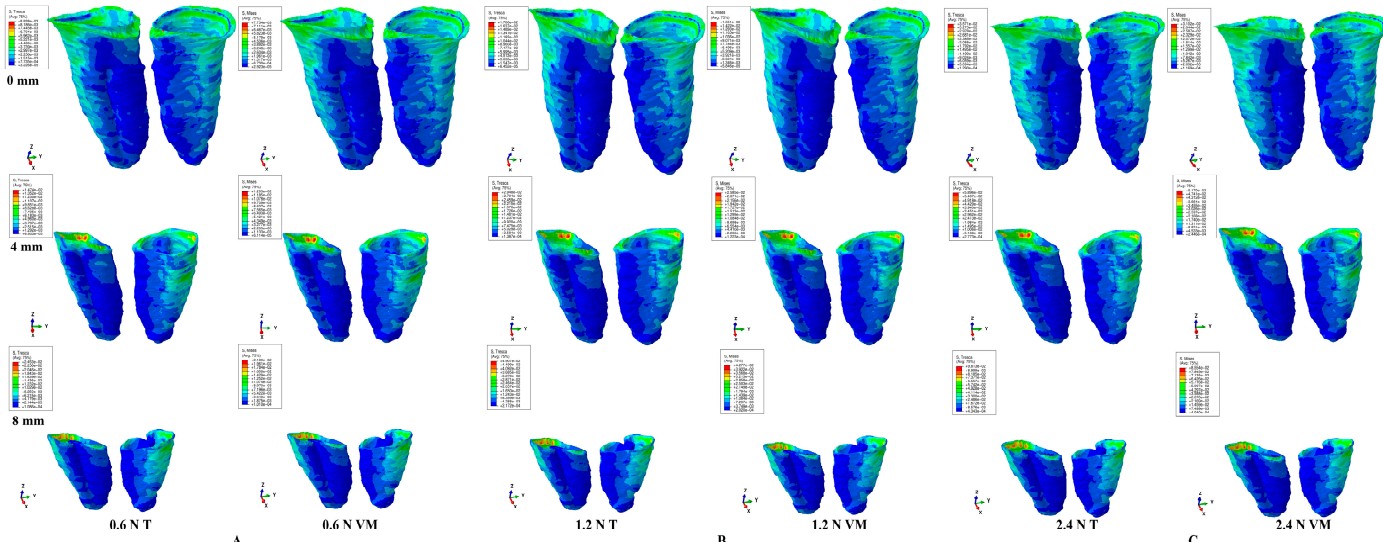

**Figure 4.** Intrusion—Tresca and VM qualitative and quantitative stress display in PDL for 0.6 N (**A**), 1.2 N (**B**), and 2.4 N (**C**), and 0-, 4-, and 8-mm periodontal breakdowns.

For rotation (Figure 5, Table 4), qualitatively, the stressed areas were in the PDL cervical third, for all three loads, all criteria, and during the entire periodontal breakdown simulation. Quantitatively, rotation displayed the highest amount of stress among all five movements. In intact periodontia experiencing up to 1.2 N, the orthodontic loads seemed safe to be applied (despite the stress in the cervical third doubling for 1.2 N). For 2.4 N, the cervical stress was up to 4.5 times higher than the MHP, suggesting a careful approach. Reduced periodontia, experiencing 0.6 N, displayed only cervical third stress, exceeding the MHP (two–four times more, after a 4 mm loss). 1.2 N displayed a moderate (up to 2.4 times) exceedance of MHP in the middle third (after a 4 mm loss) and high overrun in the cervical third (3.1–8.5 times the MHP, after 1 mm loss). 2.4 N displayed, in the entire PDL, high exceedances of MHP after 1 mm of loss. For rotation movements, more than 1.2 N should be applied with care, especially in the presence of bone loss.

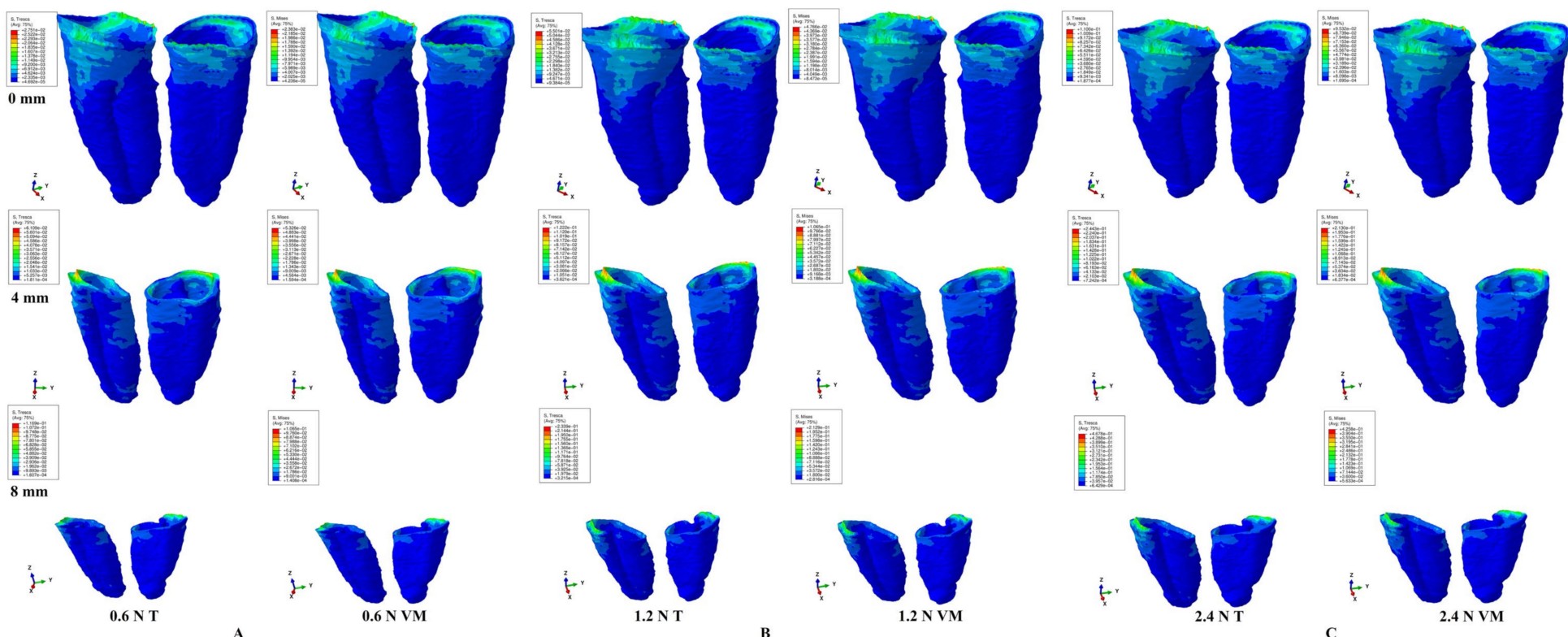

**Figure 5.** Rotation—Tresca and VM qualitative and quantitative stress display in PDL for 0.6 N (**A**), 1.2 N (**B**), and 2.4 N (**C**), and 0-, 4-, and 8-mm periodontal breakdowns.

**Table 4.** T and VM maximum stress average values (Kpa) produced by rotation in PDL.

| Resorption (mm) | | | 0 | 1 | 2 | 3 | 4 | 5 | 6 | 7 | 8 |
|---|---|---|---|---|---|---|---|---|---|---|---|---|
| Rotation 0.6 N/60 gf | T | a | 2.34 | 3.07 | 3.80 | 4.53 | 5.26 | 6.42 | 7.58 | 8.73 | 9.89 |
| | | m | 4.62 | 6.05 | 7.48 | 8.90 | 10.33 | 12.65 | 14.98 | 17.30 | **19.62** |
| | | c | **18.35** | **25.23** | **32.11** | **38.98** | **45.86** | **51.47** | **57.07** | **62.68** | **68.28** |
| | VM | a | 2.03 | 2.66 | 3.30 | 3.93 | 4.58 | 5.59 | 6.60 | 7.61 | 9.00 |
| | | m | 4.01 | 5.24 | 6.48 | 7.71 | 9.01 | 11.03 | 13.06 | 15.08 | **17.86** |
| | | c | **15.90** | **21.86** | **27.82** | **33.77** | **39.98** | **44.87** | **49.75** | **54.64** | **62.16** |
| 1.2 N/120 gf | T | a | 4.67 | 6.14 | 7.60 | 9.06 | 10.51 | 12.83 | 15.15 | 17.47 | **19.79** |
| | | m | 9.25 | 12.10 | 14.95 | **17.81** | **20.67** | **25.31** | **29.95** | **34.60** | **39.24** |
| | | c | **36.70** | **50.46** | **64.21** | **77.97** | **91.72** | **102.93** | **114.14** | **125.35** | **136.56** |
| | VM | a | 4.05 | 5.33 | 6.59 | 7.86 | 9.17 | 11.19 | 13.21 | 15.23 | **18.00** |
| | | m | 8.01 | 10.48 | 12.96 | 15.43 | **18.02** | **22.06** | **26.11** | **30.16** | **35.72** |
| | | c | **31.80** | **43.71** | **55.63** | **67.55** | **79.96** | **89.73** | **99.51** | **109.28** | **124.32** |
| 2.4 N/240 gf | T | a | 9.34 | 12.29 | 15.21 | **18.13** | **21.03** | **25.67** | **30.30** | **34.93** | **39.57** |
| | | m | **18.50** | **24.19** | **29.90** | **35.61** | **41.34** | **50.61** | **59.90** | **69.19** | **78.48** |
| | | c | **73.41** | **100.91** | **128.42** | **155.93** | **183.44** | **205.86** | **228.29** | **250.70** | **273.12** |
| | VM | a | 8.10 | 10.65 | 13.18 | 15.72 | **18.34** | **22.38** | **26.42** | **30.46** | **36.00** |
| | | m | **16.03** | **20.96** | **25.91** | **30.86** | **36.04** | **44.12** | **52.22** | **60.32** | **71.44** |
| | | c | **63.60** | **87.43** | **111.26** | **135.10** | **159.92** | **179.47** | **199.02** | **218.56** | **248.64** |
| Rotation | T/VM % | a | 1.15 | 1.15 | 1.15 | 1.15 | 1.15 | 1.15 | 1.15 | 1.15 | 1.10 |
| | | m | 1.15 | 1.15 | 1.15 | 1.15 | 1.15 | 1.15 | 1.15 | 1.15 | 1.10 |
| | | c | 1.15 | 1.15 | 1.15 | 1.15 | 1.15 | 1.15 | 1.15 | 1.15 | 1.10 |

T—Tresca, VM—Von Mises, a—apical third, m—middle third, c—cervical third, T/VM%—% increase.

For tipping (Figure 6, Table 5), qualitatively, the main stress areas were in the cervical third for all simulations, with an extension of the red–orange stress areas after 4 mm of loss. In intact periodontia, up to 1.2 N loads, no visible ischemic and resorptive risks were visible, while, for 2.4 N, stress exceeding three times the MHP was observed. Reduced periodontia under a 0.6 N load displayed a moderate stress increase, found only in the cervical third (up to three times more than the MHP for 8 mm of loss). 1.2 N displayed an extremely small stress increase in the middle third but higher stress exceedances of MHP in the cervical third (3–5.5 times higher, after 4 mm of loss). The 2.4 N load displayed higher overruns of MHP in the middle and cervical thirds, and moderated exceedances apically. Thus, after 4 mm of loss, loads higher than 1.2 N should be carefully considered.

**Table 5.** T and VM maximum stress average values (Kpa) produced by tipping in PDL.

| Resorption (mm) | | | 0 | 1 | 2 | 3 | 4 | 5 | 6 | 7 | 8 |
|---|---|---|---|---|---|---|---|---|---|---|---|---|
| Tipping 0.6 N/60 gf | T | a | 1.55 | 2.36 | 3.16 | 3.96 | 4.78 | 5.65 | 6.52 | 7.39 | 8.26 |
| | | m | 3.06 | 4.07 | 5.09 | 6.10 | 7.11 | 8.42 | 9.73 | 11.03 | 12.34 |
| | | c | 12.13 | 15.55 | **18.96** | **22.38** | **25.79** | **30.57** | **35.35** | **40.13** | **44.91** |
| | VM | a | 1.34 | 2.04 | 2.74 | 3.43 | 4.16 | 4.92 | 5.67 | 6.43 | 7.29 |
| | | m | 2.65 | 3.53 | 4.41 | 5.28 | 6.19 | 7.33 | 8.47 | 9.61 | 10.88 |
| | | c | 10.51 | 13.47 | **16.43** | 19.39 | **22.45** | 26.61 | 30.77 | 34.93 | 39.60 |
| 1.2 N/120 gf | T | a | 3.10 | 4.72 | 6.32 | 7.92 | 9.56 | 11.30 | 13.04 | 14.78 | 16.53 |
| | | m | 6.12 | 8.15 | 10.17 | 12.20 | 14.22 | **16.84** | **19.45** | **22.07** | **24.68** |
| | | c | **24.26** | **31.09** | **37.92** | **44.75** | **51.58** | **61.14** | **70.70** | **80.26** | **89.82** |
| | VM | a | 2.69 | 4.09 | 5.47 | 6.86 | 8.32 | 9.83 | 11.35 | 12.86 | 14.58 |
| | | m | 5.31 | 7.06 | 8.81 | 10.57 | 12.39 | 14.66 | **16.94** | **19.22** | **21.76** |
| | | c | **21.02** | **26.94** | **32.86** | **38.77** | **44.90** | **53.22** | **61.54** | **69.87** | **79.20** |
| 2.4 N/240 gf | T | a | 6.20 | 9.43 | 12.63 | 15.83 | **19.12** | **22.60** | **26.08** | **29.56** | **33.06** |
| | | m | 12.24 | 16.29 | **20.34** | **24.39** | **28.44** | **33.67** | **38.90** | **44.13** | **49.36** |
| | | c | **48.52** | **62.18** | **75.84** | **89.50** | **103.16** | **122.28** | **141.40** | **160.52** | **179.64** |
| | VM | a | 5.38 | 8.18 | 10.95 | 13.72 | 16.64 | **19.67** | **22.70** | **25.73** | **29.16** |
| | | m | 10.61 | 14.12 | **17.63** | **21.14** | **24.77** | **29.33** | **33.88** | **38.44** | **43.52** |
| | | c | **42.04** | **53.88** | **65.71** | **77.55** | **89.80** | **106.44** | **123.09** | **139.73** | **158.40** |
| Tipping | T/VM % | a | 1.15 | 1.15 | 1.15 | 1.15 | 1.15 | 1.15 | 1.15 | 1.15 | 1.13 |
| | | m | 1.15 | 1.15 | 1.15 | 1.15 | 1.15 | 1.15 | 1.15 | 1.15 | 1.13 |
| | | c | 1.15 | 1.15 | 1.15 | 1.15 | 1.15 | 1.15 | 1.15 | 1.15 | 1.13 |

T—Tresca, VM—Von Mises, a—apical third, m—middle third, c—cervical third, T/VM%—% increase.

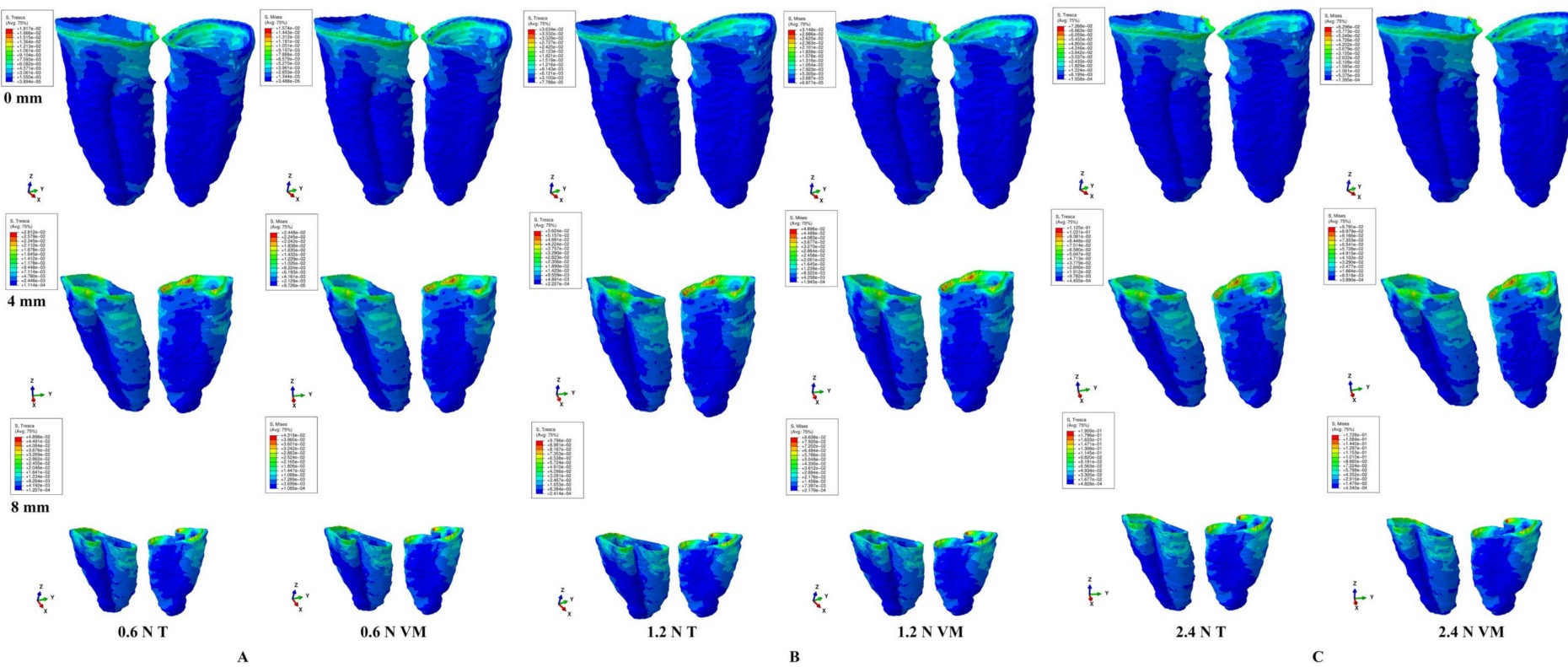

**Figure 6.** Tipping—Tresca and VM qualitative and quantitative stress display in PDL for 0.6 N (**A**), 1.2 N (**B**), and 2.4 N (**C**), and 0-, 4-, and 8-mm periodontal breakdown.

For translation (Figure 7, Table 6), qualitatively, the main stress areas were in the PDL cervical third, with a visible increase in red–orange areas after 4 mm of bone loss. Quantitatively, in intact periodontia, 0.6 N was safely applied, while the other two loads produced, on the cervical third, stress that exceeded the MHP (1.2 N—by two times, 2.4 N—by 4.3 times). In reduced periodontia, a moderate progressive stress increase (after 4 mm loss) was seen in the cervical third for 0.6 N. 1.2 N displayed a higher stress exceedance of the MHP, limited to the cervical third (3–8.4 times) and a smaller one in the middle third (up to a stress doubling at 8 mm loss). 2.4 N displayed stress in the cervical third, which was 6–16.8 times higher than the MHP, as well as a moderate stress increase in the middle and apical (mild) thirds after 4 mm of loss. Thus, a load of more than 1.2 N could significantly increase the ischemic and resorptive risks for more than 4 mm bone loss levels.

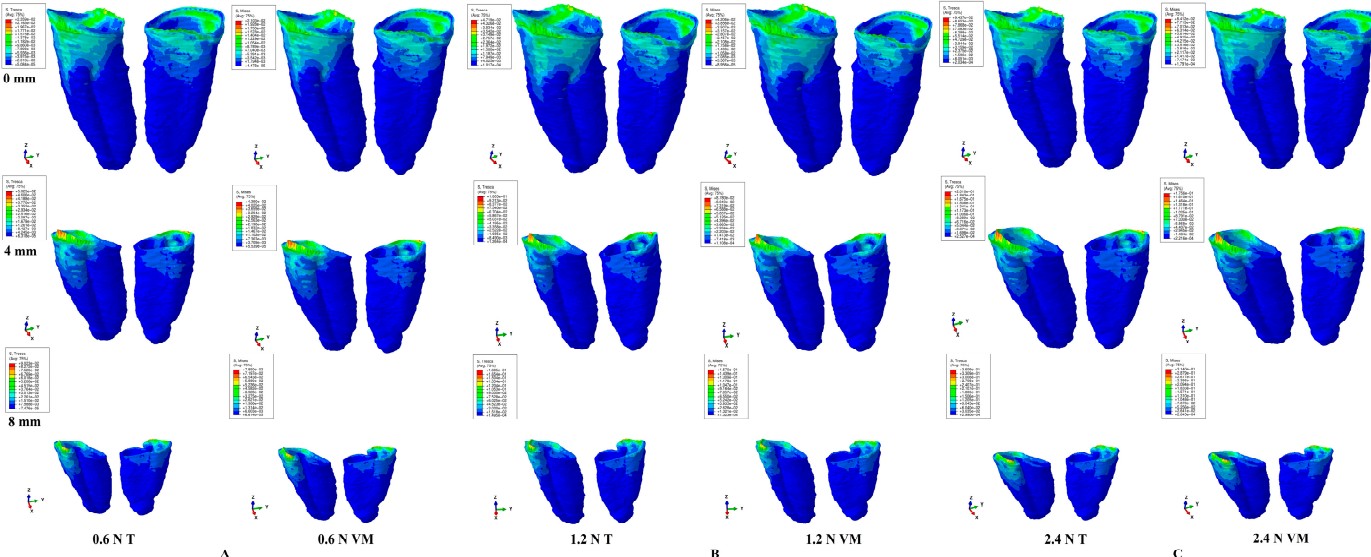

**Figure 7.** Translation—Tresca and VM qualitative and quantitative stress display in PDL for 0.6 N (**A**), 1.2 N (**B**), and 2.4 N (**C**), and 0-, 4-, and 8-mm periodontal breakdowns.

**Table 6.** T and VM maximum stress average values (KPa) produced by translation in PDL.

| Resorption (mm) | | | 0 | 1 | 2 | 3 | 4 | 5 | 6 | 7 | 8 |
|---|---|---|---|---|---|---|---|---|---|---|---|---|
| Translation 0.6 N/60 gf | T | a | 2.01 | 2.57 | 3.13 | 3.69 | 4.25 | 5.06 | 5.88 | 6.69 | 7.51 |
| | | m | 3.97 | 5.09 | 6.20 | 7.31 | 8.43 | 10.09 | 11.76 | 13.43 | 15.10 |
| | | c | 17.71 | **23.76** | **29.80** | 35.84 | 41.89 | 48.29 | 54.69 | **61.10** | **67.70** |
| | VM | a | 1.79 | 2.29 | 2.79 | 3.29 | 3.71 | 4.42 | 5.13 | 5.85 | 6.60 |
| | | m | 3.54 | 4.53 | 5.53 | 6.52 | 7.36 | 8.82 | 10.28 | 11.74 | 13.14 |
| | | c | 15.78 | **21.16** | **26.55** | 31.93 | 36.59 | 42.18 | 47.78 | 53.37 | 58.89 |
| 1.2 N/120 gf | T | a | 4.03 | 5.15 | 6.26 | 7.38 | 8.49 | 10.13 | 11.75 | 13.39 | 15.02 |
| | | m | 7.95 | 10.18 | 12.40 | 14.63 | 16.85 | **20.19** | **23.53** | **26.86** | **30.20** |
| | | c | **35.43** | **47.52** | **59.60** | **71.69** | **83.77** | 96.58 | 109.39 | 122.19 | 135.40 |
| | VM | a | 3.59 | 4.59 | 5.58 | 6.57 | 7.42 | 8.85 | 10.27 | 11.70 | 13.21 |
| | | m | 7.08 | 9.07 | 11.05 | 13.03 | 14.73 | **17.64** | **20.56** | **23.48** | **26.28** |
| | | c | **31.56** | **42.33** | **53.09** | **63.86** | **73.18** | 84.37 | 95.56 | 106.75 | 117.78 |
| 2.4 N/240 gf | T | a | 8.05 | 10.29 | 12.52 | 14.75 | 16.98 | **20.26** | **23.50** | **26.78** | **30.05** |
| | | m | 15.90 | **20.35** | **24.80** | **29.25** | **33.70** | **40.38** | **47.05** | **53.73** | **60.40** |
| | | c | **70.86** | **95.03** | **119.20** | **143.37** | **167.54** | **193.16** | **218.77** | **244.39** | **270.80** |
| | VM | a | 7.18 | 9.17 | 11.16 | 13.15 | 14.84 | **17.70** | **20.54** | **23.39** | **26.41** |
| | | m | 14.17 | **18.14** | **22.10** | **26.07** | **29.45** | **35.29** | **41.12** | **46.95** | **52.56** |
| | | c | **63.12** | **84.65** | **106.18** | **127.72** | **146.36** | **168.74** | **191.12** | **213.49** | **235.56** |
| Translation | T/VM % | a | 1.12 | 1.12 | 1.12 | 1.12 | 1.14 | 1.14 | 1.14 | 1.14 | 1.14 |
| | | m | 1.12 | 1.12 | 1.12 | 1.12 | 1.14 | 1.14 | 1.14 | 1.14 | 1.15 |
| | | c | 1.12 | 1.12 | 1.12 | 1.12 | 1.14 | 1.14 | 1.14 | 1.14 | 1.15 |

T—Tresca, VM—Von Mises, a—apical third, m—middle third, c—cervical third, T/VM%—% increase.

## 4. Discussion

Our FEA study assessed the differences between the only two failure criteria that are biomechanically suited for dental tissue numerical studies (due to their acknowledged ductile resemblance) [1]. Additionally, by approaching the boundary condition issues, our study aimed to improve the accuracy of FEA dental studies.

The present analysis was conducted on 81 3D models (with 0–8 mm bone loss) and 486 FEA simulations. This research was the first of this type to investigate the boundary conditions used in the FEA, correlated with the proper material-based failure criteria and with the MHP.

Both analyzed criteria (Tresca and VM) displayed similar qualitative stress distribution areas for all three loads, levels of bone loss, and orthodontic movements. Quantitatively significant differences were seen. For both criteria, the quantitative amounts of stress doubled (1.2 N) and quadrupled (2.4 N) when compared with 0.6 N, with Tresca being 15% higher than VM, as expected [1,21–23]. The stress and ischemic and resorptive risks increase were directly correlated with bone loss and load increase. The rotation, translation, and tipping were the most stressful movements for the PDL and the most prone to ischemia and further periodontal resorption. All quantitative stresses were lower than the maximum amount of physical properties of the tooth and the surrounding periodontium components [27–30].

In both intact and reduced periodontia, qualitatively, the rotation, translation, and tipping caused the most stress in the cervical third of the PDL (an area less vascularized than the other two thirds), while the intrusion and extrusion, despite being the least stressful, caused stress across the entire PDL (in agreement with clinical behavior). This stress distribution needs to be considered when dealing with various levels of bone loss, since, if the same applied load is kept for bone loss as for an intact periodontium, the amount of stress is higher and the ischemic and resorptive risks progressively increase. Thus, 1.2 N can be safely applied to an intact periodontium, while 2.4 N could produce mild to moderate ischemic and resorptive risks in the cervical third of the PDL (especially for the rotation, translation, and tipping movements), due to the reduced vascularization when compared with the apical third. It also must be noted that, clinically, there are rarely pure orthodontic movements (as here) and, usually (due to their combination), the amount of combined stress is lower than here.

The same load amount (0.6–1.2 N) was also safe for extrusion and intrusion for the reduced periodontium, since a smaller stress exceedance of the MHP was displayed, and only in the cervical third of the PDL. Mild to moderate ischemic risks were predictable for a 2.4 N load with more than 4 mm of loss, since the stress exceedance was seen across the entire PDL (especially in the cervical third).

For the other three more stressful movements (i.e., rotation, translation, and tipping), 1.2 N seemed to be safe for up to 4 mm of loss, whereas, after this level, moderate ischemic and resorptive risks were visible for the cervical third of the PDL (e.g., where qualitative stress display was visible). However, 2.4 N seemed to produce moderate risks for the cervical third of the PDL, especially after 4 mm of loss (due to the high exceedance of the MHP ranging from 5 to 17 times). Based on the above results, 4 mm of periodontal loss seems to be a reference point in orthodontic treatment, signaling an increase in the ischemic and resorptive risks if the same applied loads are kept as for an intact periodontium. Thus, after 4 mm of loss, to avoid any degenerative risks (i.e., pulpal and periodontal), a load of more than 1 N (approx. 100 gf) should be carefully considered, especially for movements of rotation, translation, and tipping. These results agree with Han et al.'s [36] clinical report for an intact periodontium; 0.15–0.5 N were safely applied for intrusion, while 2.25–3 N were considered severe and prone to ischemia and resorptive processes (despite clinically seeing that not all cases displayed necrosis and resorption for severe forces).

Due to the controversy regarding the optimal and maximum amounts of orthodontic load that can be safely applied to intact periodontia [3,4,9,10,13,18,25,44–46], to avoid ischemic and resorptive risks [9,10,25,31–35,47–50], the present numerical study provides new data confirming our previous reports [1,21–25] for both intact and reduced periodontia.

The quantitative difference between the Tresca and Von Mises criteria was only 15% (acceptable); thus, both criteria are safe to be used for dental tissue simulations. However, since Tresca is specially designed for non-homogenous materials (VM for homogenous) and the anatomical internal micro-architecture displays non-homogeneity, T seems to be quantitatively more suitable. Nevertheless, if only a qualitative stress display is mandatory, both criteria supply equivalent results.

The assumed boundary conditions of isotropy, linear elasticity, and perfectly bonded interfaces (i.e., with proven accuracy up to 1 N) seem to be correct up to 2.4 N, in agreement with our previous study [1]. The up-to-1 N accuracy is physically and mechanically based on the extremely small deformations and displacements that occur in the tissue's internal micro-architecture [1]. The simulations showed that the stressed areas kept their display up to 2.4 N as they did for less than 1 N forces (i.e., 0.6 N, as used here), confirming the limited deformations and displacements in the PDL. Moreover, as the PDL is the most deformable component of dental tissue, this reasoning could lead to the conclusion that all other components of dental tissue benefit from the same assumed boundary conditions as those assumed here. Regarding the perfectly bonded interfaces, it is practically impossible to introduce several types of bonding interface, since no mathematical algorithm to describe their biomechanical behavior currently exists.

In our simulation, the load distribution was uniform over the loaded area, while the amplitude was "ramp" (i.e., the loading function is linear); the load was applied in small increments up to the total amount of force. A static general analysis procedure was considered, within which the loads were not time dependent. The incremental loading scheme was employed only to prevent stress concentration, numerical problems, and premature failure. After the total amount of force was applied, the stress state remained unchanged if the force was kept constant. Moreover, dynamic analysis (time-dependent loading) is prohibitive, since it implies time-dependent material properties (which is not correct for dental tissue).

Due to the novelty of this subject, no other studies with a similar approach to this one were found. However, despite this issue, some correlations with earlier FEA-related studies [3–19] could be performed. Only Perrez et al. [2], in an older FEA study assessing the brittleness of endodontic root filling, partially approached the selection of failure criteria based on analyzed material types, but without mentioning Tresca or boundary conditions.

Most earlier FEA studies [3–19] reported, as assumed boundary conditions, isotropy, homogeneity, and linear elasticity. No mention of the type of contact between anatomical components was found. These assumed conditions were correctly employed (i.e., providing accurate results) only if the proper failure criteria were used, as our study confirmed. The easiest method to verify/validate the accuracy of results is the correlation with the MHP [1] and the acknowledged clinical biomechanical behavior [20]. Moreover, the anatomical accuracy of the analyzed PDL 3D models is also important (e.g., earlier studies employed models that were based on ideal anatomical data, with no mention of the mesh validating the means). There is a strict correlation between the failure criteria, boundary conditions, and anatomical accuracy of a FEA, to provide accurate results as in the engineering field [1]. Thus, if these correlations are not met, the FEA lacks accuracy [1]. There are many FEA studies in the current research flow; however, only a limited number qualify to be used for correlations (e.g., Roscoe et al. [5] reported only 25 from 110 examined), and none completely addressed the above-mentioned issues [3–19].

Hemanth et al. [3,4] assessed the differences in linear vs. non-linear reporting for intact PDL differences of 20–50% (maxillary central incisor, 148,097 elements and 239,666 nodes). However, despite applying 0.2–1 N intrusion and tipping, the employed failure criteria were brittle, like S1 and S3 (with plastic deformation, no recovery of the original form, and cracking/destruction), despite assessing the biomechanical behavior of a ductile-like material (with elastic deformation and recovery of the original form). Quantitatively, their stresses were higher than here (S1, 1 KPa cervically, 0.2 N intrusion; −16.4 KPa apically, 1 N of tipping/S3 −13.37 KPa apically, 0.2 N intrusion; 16.4 KPa apically, 1 N

of tipping). Thus, their result's accuracy is at least debatable, due to their use of brittle-like criteria instead of ductile-like criteria, which significantly alters the biomechanical behavior, as reported in other FEA simulations [21,22]. The employed boundary conditions were isotropy, homogeneity, and linear elasticity for all model's components (except the PDL's nonlinearity).

A new trend in PDL numerical simulations is to employ hydrostatic [6–12] pressure as a failure criterion, combined with the Ogden hyper-elastic model [5], with little correlation with the MHP. It must be emphasized that the hydrostatic pressure is a criterion specially created for liquids (with no shear stress), while the PDL's anatomical internal micro-structure, containing collagen fibers displayed as variously oriented dense fiber bundles with interposed NVB and rich circulatory vessels [22], is different from that of a liquid. Moreover, the Ogden hyper-elastic model was specially created to describe the non-linear behavior of complex materials, such as rubbers, polymers, and biological tissues (e.g., cells), and, due to PDL's anatomical internal micro-architecture, this model is not suitable. The boundary conditions employed were isotropy, linear elasticity, and homogeneity (except PDL with non-linearity). Thus, based on the above-mentioned issues, these FEA results raise a series of questions related to their results' accuracy. Moreover, Wu et al. [6–8] reported, for intact PDLs, various optimal forces (ranging 0.28–3.31 N) for canine, premolar, and lateral incisors, with significant differences found for the same tooth (e.g., canine: rotation 1.7–2.1 N [6] and 3.31 N [8]; extrusion 0.38–0.4 N [6] and 2.3–2.6 N [7]; premolar: rotation 2.8–2.9 N [8]), which were much higher than either our 0.6–1.2 N, those reported by Proffit et al. [20] (0.1–1 N), or Hemanth et al. [3,4] (0.3–1 N).

Hohmann et al. [9,10], employing the same hydrostatic pressure (same boundary conditions) on an intact PDL, reported, for 0.5–1 N intrusion, apical third stresses of 9.95e-00 TPa (vs. our 5.21–5.99 KPa for 1.2 N of intrusion). This huge amount of stress implies the destruction of the apical third, as this exceeded not only the 16 KPa of the MHP but also the dentine's maximum shear stress of 29–104 MPa [1], contradicting all clinically available data [20]. Other studies, using the same above-mentioned conditions, reported lower amounts of stress for the apical and cervical thirds of an intact PDL, i.e., 10–20 KPa [11], −5.8 KPa [5] for tipping, and −4.7 KPa [5] for intrusion, when 0.25 N were applied. Higher amounts of stress were reported by Moga et al. [22] for 0.5 N of intrusion (−13.68 KPa apically, and 18.86 KPa cervically), when performing a comparative study among five failure criteria.

Zhang et al. [12], in a recent study with the conditions of 0.1–1.5 N of intrusion, a maxillary central incisor, linear elasticity, homogeneity, and isotropy, hydrostatic stress, and S1, reported that results with assumed isotropy, homogeneity, and linear elasticity as boundary conditions were closer to clinical and animal studies, despite other studies [3–5,13] reporting the opposite. Nevertheless, hydrostatic and S1 failure criteria are not suitable for FEAs of PDL. Moreover, they reported no stress in the cervical third of the PDL for intrusion (which is biomechanically incorrect).

The Von Mises FEA studies (which were mechanically and behaviorally better for PDL [1,21–25]) also employed isotropy, homogeneity, and linear elasticity/non-linear elasticity, and reported variable results (both qualitative and quantitative) when correlated with the present results.

Toms et al. [13] investigated 1 N of extrusion in the PDL (with uniform vs. non-uniform thickness, linear vs. non-linear, VM, and S1/S3) of a lower premolar, reporting a higher amount of stress for linear non-uniform thickness (17.7 Kpa vs. 14.8 Kpa, the middle thirds), with a similar stress distribution, but with the highest stress in the apical third (vs. in the cervical third in the current study). The non-linear non-uniform PDL displayed the highest stress of 29.3 KPa apically (while measuring 0 KPa in the middle and 8.99 KPa in the cervical thirds). Our present stress distribution for extrusion displayed stress across the entire PDL, with the highest amount of stress being 11.64 KPa cervically and 5.21 KPa in the middle and apical thirds for 1.2 N, disagreeing with Toms et al.'s [13] simulation. The 1 N tipping for a non-uniform and non-linear PDL reported 6.34 Kpa of stress apically

and 6.49–8.86 KPa cervically vs. 2.69 KPa apically and 21 KPa cervically in the present study for 1.2 N. Toms et al. [13] reported differences between the linear (the cervical and middle thirds' stress) vs. non-linear (apical third stress) for ductile-like VM. The reported quantitative differences were variable, being up to 2.4 times higher for the non-linear brittle-like S1 apical stress when compared with the linear, and lower for the non-linear in the middle third. These results contradicted Hemanth et al. [3,4], Moga et al. [1,21–24], and the present distribution. However, it must be remembered that their lower premolar model consisted of only 1674 elements and 5205 nodes, artificially designed, with a sample size of one vs. the more complex models (e.g., 6.05 million here, and 239,666 nodes and 148,097 elements in Hemanth et al.'s [3,4]).

Roscoe et al. [5], in a recent systematic review study (1999–2019), assessing the constitutive models of PDL and with MHP correlations, found only 25 FEA studies of PDLs (most of them are also referenced here) from a total of 110, with only 10 assessing the correlation's quantitative results with MHP, 20 studies assuming linear elasticity, 3 studies hyper-elasticity, and 1 study non-linearity. He also analyzed an idealized premolar (VM, S1, S3, hydrostatic pressure), considering isotropy, linearity, and homogeneity for all models' components (except the PDL considered non-linear), with 1.67 million elements, and under 0.25 N–2.25 N of intrusion and tipping. There was no discussion, however, about the suitability and accuracy for the PDL study of these failure criteria from a physical-mechanical viewpoint (as above-mentioned). His 3D model was an idealized anatomical model (with a sample size of one), while our models were CBCT- and anatomical-based (with a sample size of nine). The stress display areas for VM non-linear looked vaguely like ours (i.e., the cervical third stress in the tipping, and entire PDL stress for the intrusion), while the linear simulations totally disagreed. However, it must be emphasized that Roscoe et al.'s [5] stress distribution areas are biomechanically unnatural, and not clinically met (while the present study is both mechanically and clinically feasible). The quantitative reports for the non-linear PDL are under 4.7 KPa of the MHP for 0.25 N and in exceedance for 2.4 N. His conclusion was that only hydrostatic stresses could be used in the FEA of PDLs, in total contradiction with the physical-mechanical engineering field knowledge (as above-mentioned), other FEA reports [1,21–25], and the present study.

Field et al. [14], in a model of the lower mandibular arch (incisor, canine, and premolar, with mesh size 1.2 mm, 10 nodes tetrahedral elements, 23,565 elements, 32,812 nodes, 0.35–0.5 N of tipping, VM, S1, hydrostatic stress), reported higher stress for the multiple teeth vs. the single tooth models (i.e., which is biomechanically debatable). The stress display areas (qualitative) were closer to the present study. However, the qualitative color-coded areas were entirely red–orange, signaling a quasi-uniform stress, which is biomechanically unnatural and clinically not true. Nevertheless, the applied 0.5 N of tipping was reported to produce a maximum amount of stress of 235.2–324 KPa, highly exceeding the maximum 12.8–16–26 KPa [5–8] reported for MHP, signaling high ischemic and resorptive risks for a light force, contradicting both clinical knowledge [20] and the present results.

Other two studies, namely Maravic et al. [15] (a single simplified model of the 2nd upper premolar, intrusion) and Huang et al. [16] (a single simplified model of the 1st lower premolar, intrusion), reported qualitatively comparable but quantitatively higher results. Shaw et al. [18] reported the intrusion and extrusion being more stressful than the rotation, translation, and tipping in a VM study, totally contradicting the present study and biomechanical and clinical knowledge [20]. In different VM-based research, Shetty et al. [19] reported the tipping to be more stressful than intrusion, in agreement with the results reported herein.

Gupta et al. [17] reported, for an intact PDL (specifically, an artificially created upper incisor), for intrusion (0.15 N), extrusion (0.3 N), and tipping (0.3 N), smaller amounts of stress than here and maximum stress areas in the apical third (which is biomechanically unnatural for such small forces), contradicting our results.

The above-mentioned correlations with available studies [3–19] clearly show the main shortcomings of applying the FEA method for the study of dental tissue, and why it lacks the accuracy shown in the engineering field. Those reported above are due to significant differences between the analyzed 3D models (i.e., mostly idealized, with low anatomical accuracy due to a high global element size of 1.2 mm and fewer elements/nodes 1674/5205–23,563/32,812, up to 1.67 million, vs. a reduced global element size of 0.08–0.116 mm and a higher number of elements/nodes, specifically 5.06–6.05 million/ 0.97–1.07 million, in the present study). Most of these studies do not motivate their selection of their employed failure criteria (material type-based), instead employing, comparing, and correlating brittle-like or liquid criteria, which are not suited for the ductile-like nature of dental tissue. Thus, it is expected that the reported stress distribution (qualitatively) varies and displays unnatural or unrealistic clinical results, as proven in our earlier comparative studies [1,21–25]. Their quantitative results show a high exceedance of the MHP, signaling ischemia and resorptive processes that clinically do not appear [20]. The boundary conditions assumed are largely isotropy, homogeneity, and linear elasticity, while only a small number of studies considered non-linearity for the PDL and reported contradictory results.

FEA has, as a main limitation, the fact that it cannot accurately reproduce clinical conditions; however, it is the only available method that enables the individual study of each tissular component, supplying data that cannot be otherwise obtained [1,21–25]. The biomechanical behavior of dental tissue, and especially of the PDL, is extremely complex, due to the internal tissular micro-architecture (which cannot be accurately numerically modeled) and the clinical combination of various orthodontic movements (only pure movements are used in current available FEAs), as well as ways to transfer the orthodontic load. The fixed appliance (the stainless steel bracket) clinically induces multiple forces and ways of transfer that cannot be simulated and quantified. No dental FEA could accurately reproduce the clinical behavior of dental tissue, and, thus, the results should be correlated with clinical knowledge (clinical biomechanical behavior and maximum hydrostatic pressure) [1,21–25]. We expected these issues and tried to compensate for these shortcomings through numerical data interpretation (especially those close to physiological limit) and force appliance on the bracket base. The present study followed the same methodology as previous FEA dental studies (mentioned in references [3–19]), but only using the correct material-based failure criteria and the same assumed isotropy, homogeneity/non-homogeneity, and linear elasticity boundary conditions, as the previous numerical studies did. However, the present study, by correctly using the above-mentioned conditions, obtained correct qualitative and quantitative results (indirectly validated by correlation with maximum hydrostatic pressure and clinical data). The aim of the present study was to verify a set of rules to ensure FEA provided correct results that were in agreement with clinical data (which is a progression in the application of FEA to dentistry), since the previous analyses used a mix of failure criteria and boundary conditions that provided clinically contradictory and incorrect results. In numerical studies, by changing the parameter, a complete new set of results can be obtained [1,21–25]. Thus, since the input data can be easily changed, the required sample size is extremely small (e.g., the sample sizes of one in the above-mentioned FEA studies [3–19] vs. nine herein), thus allowing multiple simulations with various results from a small number of patients [1,21–25]. FEAs are descriptive studies [3–19], fundamentally different from the clinical ones (with a different set of rules and requirements, the sample size being one of them [1,21–25]), which is clearly visible in the numerical dental study methodologies in the current research flow (i.e., a sample size of one—one patient/model and few simulations [3–19] vs. the nine patients, 81 models, and 486 simulations herein). The analysis herein has a sample size nine times higher than most FEA studies [3–19], since we agree that more patients and models imply more simulations and data that produce better results and conclusions [1,21–25]. To verify the accuracy of the above-mentioned boundary conditions and failure criteria, more numerical simulations are needed, to improve both the modeling (i.e., internal micro-architecture) technique and the boundary condition assumptions (linear vs. non-linear and anisotropy), in order to enhance the accuracy and transform FEAs in

dentistry into a reliable everyday study method. The accuracy of numerical simulations can be easily improved in the dental field by following the requirements set out in the engineering field and by introducing the help of new artificial intelligence (AI) (due to the complexity of non-linear equations).

**5. Conclusions**

1.  Both Tresca and Von Mises failure criteria displayed similar qualitative results, while the quantitative ones were 15% higher for T. Since the Tresca criteria were designed to describe non-homogenous materials' behavior, they are better suited for quantitative results than VM and should therefore be preferably used for FEA simulations.
2.  For all three loads, the qualitative stress display was similar for both failure criteria and for all bone loss levels, implying that the deformations and displacements were constant, and manifested in the same areas independently of the load's amount, the only difference being their intensity (stress was doubled for 1.2 N and quadrupled for 2.4 N).
3.  The boundary condition assumptions (homogeneity, linear elasticity, isotropy, and perfectly bonded interfaces) seem to be correct for up to 2.4 N of an applied load, and thus correctly used in the FEA of dental tissues.
4.  Clinically speaking, in an intact periodontium, forces up to 2.4 N could be relatively safely applied (with mild risks), whereas, in a reduced periodontium after 4 mm of loss, any more than 1 N should be carefully considered (with high ischemic and resorptive risks).
5.  Clinically speaking, 4 mm of bone loss seems to be a reference point in orthodontic treatment, since, from this moment, ischemic and resorptive risks significantly increase, and the benefits must be balanced with risks.

**Author Contributions:** Conceptualization: R.-A.M.; methodology: R.-A.M.; software: R.-A.M.; validation: R.-A.M. and C.D.O.; formal analysis: R.-A.M.; investigation: R.-A.M.; resources: R.-A.M.; data curation: R.-A.M.; writing—original draft preparation: R.-A.M.; writing—review and editing: R.-A.M., A.G.D. and C.D.O.; visualization, supervision, and project administration: R.-A.M., A.G.D. and C.D.O.; funding acquisition: R.-A.M., A.G.D. and C.D.O. All authors have read and agreed to the published version of the manuscript.

**Funding:** The authors were the funders of this research project.

**Institutional Review Board Statement:** The research protocol was approved by the Ethical Committee of the University of Medicine (158/2 April 2018).

**Informed Consent Statement:** Informed oral consent was obtained from all subjects involved in the study.

**Data Availability Statement:** The raw data supporting the conclusions of this article will be made available by the authors on request.

**Conflicts of Interest:** The authors declare that they have no conflicts of interest.

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
