# Peer review of "The Importance of Boundary Conditions and Failure Criterion in Finite Element Analysis Accuracy—A Comparative Assessment of Periodontal Ligament Biomechanical Behavior"

_applsci, doi:10.3390/app14083370_

Round 1

Reviewer 1 Report

Comments and Suggestions for Authors

This article seems to me well constructed and well presented. the results are quite clear.  it would be nice if in future studies a hint or reference were made to the clinical application of these results in fixed or clear aligners orthodontics by correlating it with the forces that can be used.

Please indicate which type of article was conducted in the header of the page

Author Response

Department of Cariology, Endodontics and Oral Pathology

School of Dental Medicine

University of Medicine and Pharmacy

Ms. Betty Pei

Section Managing Editor

Applied Sciences-Applied Dentistry and Oral Sciences      

Special Issue - Oral Diseases and Clinical Dentistry      

                                                                                                                               March 5th, 2024

Dear Ms. Betty Pei,

Thank you very much for your letter dated February 27th, 2024, with the comments of the reviewers. We have now carefully considered the comments of the reviewers and amended the paper accordingly. All changes are highlighted in red throughout the manuscript and included also below.

Reply to Reviewer #1:

We agree and we thank the reviewer for his/her time and comments. Appropriate changes in the manuscript have by now been made. Please see below and in the manuscript.

Concern of the reviewer:

” Comments and Suggestions for Authors

This article seems to me well constructed and well presented. the results are quite clear.  it would be nice if in future studies a hint or reference were made to the clinical application of these results in fixed or clear aligners orthodontics by correlating it with the forces that can be used.

Please indicate which type of article was conducted in the header of the page”

Point-by-point response to the reviewer’s comments:

  1. Concern of the reviewer:

“it would be nice if in future studies a hint or reference were made to the clinical application of these results in fixed or clear aligners orthodontics by correlating it with the forces that can be used”

Our response:

  • We thank the reviewer for his/her concern and comments. We do hope that our changes are according to the reviewer‘s remarks.

We would like to address this issue in a future study. It must be emphasized that our research is a step-by-step study, thus new directions of development are always welcomed.

  1. Concern of the reviewer:

“Please indicate which type of article was conducted in the header of the page”

Our response:

  • We thank the reviewer for his/her concern and comments. We do hope that our changes are according to the reviewer‘s remarks.

Revised text: pg.1 line 1.

“Type of the Paper (Article)”

Reviewer 2 Report

Comments and Suggestions for Authors

Thank you for the opportunity to review this paper.

I have some suggestions:

Line 1 – please – choose the type of your manuscript.

Line 4 – I would suggest not to use abbreviations in the title. Please provide the full name instead of the abbreviation.

Line 143-145 – The inclusion criteria allow you to create a study group from some larger initial group. The next step is to apply the exclusion criteria - these apply to this preselected group based on the inclusion criteria, not the entire initial group. Thus, the "excluding ones were the opposite" indicated by the authors does not apply, because then all those who were included in the first stage of group formation would have to be excluded. On the other way, as the inclusion criteria was “no malposition” and also “orthodontic treatment indication” - it is mutually exclusive. Usually as indication to orthodontic treatment is tooth malposition. Have the examined patients ever received orthodontic treatment before? In the age almost 30 years they may have received treatment during their childhood.

Line 213 – “No visible influence of age, sex or gender was seen.” – please correct this unnecessary repetition.

The research looks interesting and indicative of the effect of various types of orthodontic force on the condition of periodontal tissues. I am not an expert in physics, however, one would have to look at the issue of the application of force moving the tooth. The shape of the bracket alone may not be enough. Today, displacement is caused by brackets with different types of slots, different shapes of arches (in cross-section), brackets both bonded on the vestibular side and on the palatal or lingual side, as well as by different shaped attachments in alliner treatment. These issues are worth including additionally in the discussion as an indication of the limitations of this study.

Author Response

Department of Cariology, Endodontics and Oral Pathology

School of Dental Medicine

University of Medicine and Pharmacy

Ms. Betty Pei

Section Managing Editor

Applied Sciences-Applied Dentistry and Oral Sciences      

Special Issue - Oral Diseases and Clinical Dentistry      

                                                                                                                               March 5th, 2024

Dear Ms. Betty Pei,

Thank you very much for your letter dated February 27th, 2024, with the comments of the reviewers. We have now carefully considered the comments of the reviewers and amended the paper accordingly. All changes are highlighted in red throughout the manuscript and included also below.

Reply to Reviewer #2:

We agree and we thank the reviewer for his/her time and comments. Appropriate changes in the manuscript have by now been made. Please see below and in the manuscript.

Concern of the reviewer:

” I have some suggestions:

Line 1 – please – choose the type of your manuscript.

Line 4 – I would suggest not to use abbreviations in the title. Please provide the full name instead of the abbreviation.

Line 143-145 – The inclusion criteria allow you to create a study group from some larger initial group. The next step is to apply the exclusion criteria - these apply to this preselected group based on the inclusion criteria, not the entire initial group. Thus, the "excluding ones were the opposite" indicated by the authors does not apply, because then all those who were included in the first stage of group formation would have to be excluded. On the other way, as the inclusion criteria was “no malposition” and also “orthodontic treatment indication” - it is mutually exclusive. Usually as indication to orthodontic treatment is tooth malposition. Have the examined patients ever received orthodontic treatment before? In the age almost 30 years they may have received treatment during their childhood.

Line 213 – “No visible influence of age, sex or gender was seen.” – please correct this unnecessary repetition.

The research looks interesting and indicative of the effect of various types of orthodontic force on the condition of periodontal tissues. I am not an expert in physics, however, one would have to look at the issue of the application of force moving the tooth. The shape of the bracket alone may not be enough. Today, displacement is caused by brackets with different types of slots, different shapes of arches (in cross-section), brackets both bonded on the vestibular side and on the palatal or lingual side, as well as by different shaped attachments in alliner treatment. These issues are worth including additionally in the discussion as an indication of the limitations of this study."

Point-by-point response to the reviewer’s comments:

  1. Concern of the reviewer:

“Line 1 – please – choose the type of your manuscript.”

Our response:

  • We thank the reviewer for his/her concern and comments. We do hope that our changes are according to the reviewer‘s remarks.

Revised text: pg.1 lines 1

“Type of the Paper (Article)”

  1. Concern of the reviewer:

“Line 4 – I would suggest not to use abbreviations in the title. Please provide the full name instead of the abbreviation..”

Our response:

  • We thank the reviewer for his/her concern and comments. We do hope that our changes are according to the reviewer‘s remarks.

Revised text: pg.1 lines 2-4.

The Importance of Boundary Conditions and Failure Criterion in Finite Elements Analysis Accuracy - A Comparative Assessment of Periodontal Ligament Biomechanical Behavior

  1. Concern of the reviewer:

“Line 143-145 – The inclusion criteria allow you to create a study group from some larger initial group. The next step is to apply the exclusion criteria - these apply to this preselected group based on the inclusion criteria, not the entire initial group. Thus, the "excluding ones were the opposite" indicated by the authors does not apply, because then all those who were included in the first stage of group formation would have to be excluded. On the other way, as the inclusion criteria was “no malposition” and also “orthodontic treatment indication” - it is mutually exclusive. Usually as indication to orthodontic treatment is tooth malposition. Have the examined patients ever received orthodontic treatment before? In the age almost 30 years they may have received treatment during their childhood.”

Our response:

  • We thank the reviewer for his/her concern and comments. We do hope that our changes are according to the reviewer‘s remarks.

The reviewer has a point. We corrected the text. The patients had orthodontic treatment indication, but none of them had orthodontic treatment in childhood. We were interested in the studied region of the mandibular arch to be complete (teeth and reduced bone loss), since we needed to obtain intact periodontium models.

Revised text: pg.4 lines:160-165

“The including criteria were complete mandibular dental arch in region of interest, with no malposition and intact teeth in the analyzed region, no advanced bone loss, non-inflamed periodontium, orthodontic treatment indication, proper oral hygiene. The exclusion criteria were related to the incomplete mandibular arch, malposition, non-intact teeth, in the region of interest.”

  1. Concern of the reviewer:

“Line 213 – “No visible influence of age, sex or gender was seen.” – please correct this unnecessary repetition.”

Our response:

  • We thank the reviewer for his/her concern and comments. We do hope that our changes are according to the reviewer‘s remarks.

We corrected.

Revised text: pg.6 lines 241-244

“Herein FEA analysis totaled 486 simulations displaying qualitative and quantitative results for PDL for both criteria.

The qualitative results displayed similar stress distribution for both criterions. There were no differences in stress display among the three loads.”

  1. Concern of the reviewer:

“The research looks interesting and indicative of the effect of various types of orthodontic force on the condition of periodontal tissues. I am not an expert in physics, however, one would have to look at the issue of the application of force moving the tooth. The shape of the bracket alone may not be enough. Today, displacement is caused by brackets with different types of slots, different shapes of arches (in cross-section), brackets both bonded on the vestibular side and on the palatal or lingual side, as well as by different shaped attachments in alliner treatment. These issues are worth including additionally in the discussion as an indication of the limitations of this study.”

Our response:

  • We thank the reviewer for his/her concern and comments. We do hope that our changes are according to the reviewer‘s remarks.

The reviewer has a point regarding the multiple ways to transfer the forces and determine the orthodontic movements. This is why we tried to compensate by applying the forces directly to the bracket base.

Revised text: pg.16 lines 561-5582

“The biomechanical behavior of dental tissues and especially of PDL is extremely complex since the internal tissular micro-architecture (that cannot be accurately numerically modeled) and clinical combination of various orthodontic movements (only pure movements are used in current available FEA analysis) as well as ways to transfer the orthodontic load. The fixed appliance (stainless steel-bracket) clinically induces multiple forces and ways of transfer that cannot be simulated and quantified. No dental FEA analysis could accurately reproduce the clinical behavior of dental tissues, and thus the results should be correlated with clinical knowledge (clinical biomechanical behavior and maximum hydrostatic pressure). We expected these issues and tried to compensate for these shortcomings through numerical data interpretation (especially those close to physiological limit) and force appliance on the bracket base. Herein study followed the same methodology as previous FEA dental studies (mentioned in references), with only using the correct material-based type failure criteria and the same assumed isotropy, homogeneity/non-homogeneity and linear-elasticity as boundary conditions as did the previous numerical studies. However, herein study by correctly using the above-mentioned obtained correct qualitative and quantitative results (indirectly validated by correlation with maximum hydrostatic pressure and clinical data). The aim of herein study was to verify a set of rules to make the FEA analysis to provide correct results that are in agreement with clinical data (which is a progress in FEA applied in dentistry), since the previous analysis use a mix of failure criteria and boundary condition that provided clinically contradictory and incorrect results.”

Reviewer 3 Report

Comments and Suggestions for Authors

1.      The title of the article does not express the aim of this research. The objective of the work is not clear because it talks about some boundary conditions that are not clear to carry out the analysis.

2.      The Introduction section can be more concrete. For example, the authors should consider highlighting the boundary conditions, material properties, and failure criteria of dental tissue.

3.      The main area of improvement for the current presentation is to demonstrate the use of this research.

4.       The actual biomechanics variables are simplified. Could you mind adding a figure or a boundary conditions diagram about how to be considered in the dental tissue?

5.      The case studies presented in this article are not clearly explained. Also, the numerical analysis used to evaluate the cases of study should be in dynamic behavior considered chewing force and with dentine anisotropic properties.

6.      The boundary conditions specify how the solution of the problem behaves at the boundaries of the domain. This is crucial to correctly define the problem being addressed and ensure that the solution is meaningful and applicable to the given context. For this reason, proper boundary conditions can help ensure the numerical stability of methods used to solve partial differential equations or other mathematical problems. Inadequate conditions can lead to numerically unstable or divergent solutions. Also, Boundary conditions should reflect the actual physical conditions of the problem being modeled. For example, in teeth tissue, the boundary conditions could represent fixed gum tissue edges and specified chewing forces. Would you mind explaining in depth the numerical stability of all the cases of study that have been developed in this research, the physical consistency, the mesh stability, and the Solution accuracy? Could you summarize it in a table?

7.      Really, you have done the FEM mesh with all stl gaps. I assume that the size of finite elements are too small. For that reason, the time solutions depend on the computer used. Could You please mention the computer that was used to resolve those cases of study?

8. How was it used in this research? If any biomechanical considerations have not been mentioned. Why?

9. The conclusion needs to summarize the significant findings of methods used with future proposals.

Comments on the Quality of English Language

no comments

Author Response

Department of Cariology, Endodontics and Oral Pathology

School of Dental Medicine

University of Medicine and Pharmacy

Ms. Betty Pei

Section Managing Editor

Applied Sciences-Applied Dentistry and Oral Sciences      

Special Issue - Oral Diseases and Clinical Dentistry      

                                                                                                                          March 5th, 2024

Dear Ms. Betty Pei,

Thank you very much for your letter dated February 27th, 2024, with the comments of the reviewers. We have now carefully considered the comments of the reviewers and amended the paper accordingly. All changes are highlighted in red throughout the manuscript and included also below.

Reply to Reviewer #3:

We agree and we thank the reviewer for his/her time and comments. Appropriate changes in the manuscript have by now been made. Please see below and in the manuscript.

Concern of the reviewer:

” Comments and Suggestions for Authors

The title of the article does not express the aim of this research. The objective of the work is not clear because it talks about some boundary conditions that are not clear to carry out the analysis.

  1. The Introduction section can be more concrete. For example, the authors should consider highlighting the boundary conditions, material properties, and failure criteria of dental tissue.
  2. The main area of improvement for the current presentation is to demonstrate the use of this research.
  3.  The actual biomechanics variables are simplified. Could you mind adding a figure or a boundary conditions diagram about how to be considered in the dental tissue?
  4. The case studies presented in this article are not clearly explained. Also, the numerical analysis used to evaluate the cases of study should be in dynamic behavior considered chewing force and with dentine anisotropic properties.
  5. The boundary conditions specify how the solution of the problem behaves at the boundaries of the domain. This is crucial to correctly define the problem being addressed and ensure that the solution is meaningful and applicable to the given context. For this reason, proper boundary conditions can help ensure the numerical stability of methods used to solve partial differential equations or other mathematical problems. Inadequate conditions can lead to numerically unstable or divergent solutions. Also, Boundary conditions should reflect the actual physical conditions of the problem being modeled. For example, in teeth tissue, the boundary conditions could represent fixed gum tissue edges and specified chewing forces. Would you mind explaining in depth the numerical stability of all the cases of study that have been developed in this research, the physical consistency, the mesh stability, and the Solution accuracy? Could you summarize it in a table?
  6. Really, you have done the FEM mesh with all stl gaps. I assume that the size of finite elements are too small. For that reason, the time solutions depend on the computer used. Could You please mention the computer that was used to resolve those cases of study?
  7. How was it used in this research? If any biomechanical considerations have not been mentioned. Why?
  8. The conclusion needs to summarize the significant findings of methods used with future proposals..”

Point-by-point response to the reviewer’s comments:

  1. Concern of the reviewer:

“The title of the article does not express the aim of this research. The objective of the work is not clear because it talks about some boundary conditions that are not clear to carry out the analysis.”

Our response:

  • We thank the reviewer for his/her concern and comments. We do hope that our changes are according to the reviewer‘s remarks.

The title of the article is “The Importance of Boundary Conditions and Failure Criterion in Finite Elements Analysis Accuracy - A Comparative Assessment of Periodontal Ligament Biomechanical Behavior”

Our article assessed the use of linear-elasticity, isotropy, and homogeneity/non-homogeneity as boundary conditions as well as the failure criteria selection for ductile materials for PDL study. We chose this title since there is a lack of knowledge regarding the use/employment of the above mentioned in the FEA study of dental tissues. Our study is a continuation of a larger research that observed the employment of various boundary conditions and failure criteria without any mention of the logical/mechanical reason and since the results contradicted the clinical data, varied from one study to another casting a doubt to the correctness of FEA in dentistry.

Revised text: pg.2 lines 61-79

 “The FEA dental studies [3-19], despite their considerable number, supplies various results that differ from one study to another and sometimes contradict clinical data [6-8, 20]. If the engineering field benefits from a long-time knowledge of material’s physical properties and behavioral types, the dental field lacks all these [1].

FEA begins with selection of the proper failure criteria (mathematical algorithm describing material’s behavioral way as brittle, ductile or liquid/gas). Ductile materials have a variable ability to deform and recover when subjected to a load, while brittle materials deform little, usually do not recover from deformations, and suffer from cracks and destruction. The dental tissues are considered ductile resemblance materials (with a certain brittle flow mode) [1, 21-24]. The ductile materials failure criteria are Tresca (T, maximum shear stress) and Von Mises (VM, maximum overall stress), brittle criterions are S1 (maximum principal stress) and S3 (minimum principal stress), while for liquids Hydrostatic pressure is mandatory [21, 22]. In each of these criteria the variable boundary conditions are the initial parameters that help solving the differential equations and study the biomechanical behavior under specific physical properties (linear-elasticity, homogeneity and isotropy vs. non-linear elasticity, non-homogeneity, and anisotropy). The results are also significantly influenced by the analyzed 3D structural model (i.e., anatomical accuracy), obtained either by CBCT’s anatomical reconstruction (high accuracy) or artificially created based on anatomical data (low accuracy) [2-4, 6-11, 14-16, 18, 19].”

Revised text: pg.2 lines 91-102

“Anatomically and biomechanically dental tissues are non-homogenous, anisotropic and do not show linear elasticity [23]. However, all dental studies (PDL included) neglected these issues assuming linear-elasticity, homogeneity, and isotropy (as boundary conditions assumptions), obeying Hook’s law, due to easier mathematical equations and lack of awareness. Nevertheless, from the mechanical point of view (common engineering knowledge), up to 1 N of load due to small deformations and displacements all tissues assume linear-elasticity and isotropy [13, 21-24]. However, for more than 1 N there is no available data about the linear elasticity and isotropy assumptions. Regarding the non-homogeneity/homogeneity issue of dental tissues, Tresca criterion is suited for ductile non-homogenous, while VM for ductile homogenous, as correctly reported earlier analysis [1, 21-24].”

Revised text: pg.3 lines 142-150

“Herein aims were a) to assess qualitative and quantitative differences between T (non-homogenous) and VM (homogenous) failure criterion by simulating 0-8 mm periodontal breakdown under five orthodontic movements (extrusion, intrusion, rotation, tipping and translation) and three loads (0.6, 1.2 and 2.4 N) in PDL, b) to verify the use of linear-elasticity, isotropy, homogeneity/non-homogeneity as boundary conditions assumptions for more than 1 N loads for PDL, c) to correlate the quantitative results with MHP and available clinical knowledge, evaluating ischemic and resorptive risks for more than 1 N orthodontic loads in PDL.”

  1. Concern of the reviewer:

“2.      The Introduction section can be more concrete. For example, the authors should consider highlighting the boundary conditions, material properties, and failure criteria of dental tissue.”

Our response:

  • We thank the reviewer for his/her concern and comments. We do hope that our changes are according to the reviewer‘s remarks.

Revised text: pg.2 lines 61-79

“The FEA dental studies [3-19], despite their considerable number, supplies various results that differ from one study to another and sometimes contradict clinical data [6-8, 20]. If the engineering field benefits from a long-time knowledge of material’s physical properties and behavioral types, the dental field lacks all these [1].

 FEA begins with selection of the proper failure criteria (mathematical algorithm describing material’s behavioral way as brittle, ductile or liquid/gas). Ductile materials have a variable ability to deform and recover when subjected to a load, while brittle materials deform little, usually do not recover from deformations, and suffer from cracks and destruction. The dental tissues are considered ductile resemblance materials (with a certain brittle flow mode) [1, 21-24]. The ductile materials failure criteria are Tresca (T, maximum shear stress) and Von Mises (VM, maximum overall stress), brittle criterions are S1 (maximum principal stress) and S3 (minimum principal stress), while for liquids Hydrostatic pressure is mandatory [21, 22]. In each of these criteria the variable boundary conditions are the initial parameters that help solving the differential equations and study the biomechanical behavior under specific physical properties (linear-elasticity, homogeneity and isotropy vs. non-linear elasticity, non-homogeneity, and anisotropy). The results are also significantly influenced by the analyzed 3D structural model (i.e., anatomical accuracy), obtained either by CBCT’s anatomical reconstruction (high accuracy) or artificially created based on anatomical data (low accuracy) [2-4, 6-11, 14-16, 18, 19].”

Revised text: pg.2 lines 92-102

“Anatomically and biomechanically dental tissues are non-homogenous, anisotropic and do not show linear elasticity [23]. However, all dental studies (PDL included) neglected these issues assuming linear-elasticity, homogeneity, and isotropy (as boundary conditions assumptions), obeying Hook’s law, due to easier mathematical equations and lack of awareness. Nevertheless, from the mechanical point of view (common engineering knowledge), up to 1 N of load due to small deformations and displacements all tissues assume linear-elasticity and isotropy [13, 21-24]. However, for more than 1 N there is no available data about the linear elasticity and isotropy assumptions. Regarding the non-homogeneity/homogeneity issue of dental tissues, Tresca criterion is suited for ductile non-homogenous, while VM for ductile homogenous, as correctly reported earlier analysis [1, 21-24].”

Revised text: pg.3 lines 109-115.

“The acknowledged dental components’ physical properties: cortical bone - 16.7 GPa of compressive modulus, and 157 MPa of compressive strength; trabecular/cancellous bone 0.155 GPa of compressive modulus, and 6 MPa of compressive strength; enamel - 62.2 MPa of compressive stress, 11.5-42.1 MPa of maximum tensile strength, and 53.9-104 MPa of maximum shear stress; dentine – 29-73.1 MPa of maximum shear stress, and enamel-dentine 53.9-104 MPa of maximum shear stress; PDL maximum tolerable stress 15-26KPa [1, 21-24].”

  1. Concern of the reviewer:

“3.      The main area of improvement for the current presentation is to demonstrate the use of this research.”

Our response:

  • We thank the reviewer for his/her concern and comments. We do hope that our changes are according to the reviewer‘s remarks.

Herein research, which is the single study of this type, has two main uses: clinically and theoretically (for research purpose).

From the clinical point of view, it provides the maximum amount of orthodontic force that could be applied in intact and reduced periodontium for each orthodontic movement without the risk of ischemia and further periodontal loss. It provided both quantitative (correlated with maximum hydrostatic pressure) and qualitative (stress distribution areas) results, both in agreement with the clinical knowledge. For being able to do this both failure criteria (for ductile resemblance materials) were used.

Revised text: pg. 1 lines 13-20

For a clinician knowing the amounts of load that can be safely applied during the periodontal breakdown helps in improving the predictability of the orthodontic treatment and avoiding the ischemic and resorptive risks. Thus, knowing that intact periodontium could bear up to 2.4 N without major ischemic or resorptive risks is of extreme importance. The 4 mm breakdown reference point, after which the applied loads should be lower than 1 N, supplies valuable data for both orthodontics and periodontology. The stress distribution areas displayed for each movement and bone loss level create a general complete image of PDL biomechanical behavior.

From the theoretical point of view, it provides evidence regarding the correct use of FEA in dentistry for obtaining accurate results. Thus, the use of ductile material type failure criteria (Tresca and Von Mises), and boundary conditions (isotropy, linear-elasticity and homogeneity/non-homogeneity), known to be correct up to 1 N of orthodontic loads, are proven to be correct and provide accurate results up to 2.4 N (in agreement with clinical knowledge). This issue is of extreme importance since the current available numerical studies (due to lack of awareness regarding the use of FEA in dentistry) provided variable results from one study to another and frequently contradicted the clinical knowledge (as our study mentioned).

Revised text: pg.1 lines:21-29

For a researcher providing a way to gain the much-needed results accuracy for dental studies comparable with those provided by the engineering field is valuable, since FEA is the only available method that allows individual study of each dental tissues’ component and the current numerical studies produced debatable and sometimes contradictory results. Thus, by employing the ductile resemblance material type failure criteria (T and VM), and linear-elasticity, isotropy, and homogeneity/non-homogeneity as boundary conditions assumptions in the study of PDL, herein study obtained results which are in agreement with clinical knowledge. Moreover, the above-mentioned boundary conditions are correct up to an applied load of 2.4 N (since up to 1 N are acknowledged to be mechanically correct).

  1. Concern of the reviewer:

“4.       The actual biomechanics variables are simplified. Could you mind adding a figure or a boundary conditions diagram about how to be considered in the dental tissue?”

Our response:

  • We thank the reviewer for his/her concern and comments. We do hope that our changes are according to the reviewer‘s remarks.

The biomechanical behavior of dental tissues and especially of PDL is extremely complex since the internal tissular micro-architecture (that cannot be accurately numerically modeled) and combination of various orthodontic movements (only pure movements are used in current available FEA analysis). The fixed appliance (stainless steel-bracket) also induces multiple forces that cannot be simulated and quantified. No dental FEA analysis could accurately reproduce the clinical behavior of dental tissues, and thus the results should be correlated with clinical knowledge regarding the clinical biomechanical behavior and the maximum hydrostatic pressure. Herein study followed the same methodology as previous FEA dental studies (also mentioned in references), with only using the correct material-based type failure criteria and the same assumed isotropy, homogeneity/non-homogeneity and linear-elasticity as boundary conditions as did the previous numerical studies. However, herein study by correctly using the above-mentioned obtained correct qualitative and quantitative results (indirectly validated by correlation with maximum hydrostatic pressure and clinical data). The aim of herein study was to obtain a set of rules/a way to make the FEA analysis to provide correct results that are in agreement with clinical data, since the previous analysis use a mix of failure criteria and boundary condition that provided clinically contradictory results.

Revised text: pg.16 lines 558-582

“The FEA analysis has as a main limitation the fact that it cannot accurately reproduce the clinically conditions, however it is the only available method that enables the individual study of each tissular component supplying data that cannot be otherwise obtained. The biomechanical behavior of dental tissues and especially of PDL is extremely complex since the internal tissular micro-architecture (that cannot be accurately numerically modeled) and clinical combination of various orthodontic movements (only pure movements are used in current available FEA analysis) as well as ways to transfer the orthodontic load. The fixed appliance (stainless steel-bracket) clinically induces multiple forces and ways of transfer that cannot be simulated and quantified. No dental FEA analysis could accurately reproduce the clinical behavior of dental tissues, and thus the results should be correlated with clinical knowledge (clinical biomechanical behavior and maximum hydrostatic pressure). We expected these issues and tried to compensate for these shortcomings through numerical data interpretation (especially those close to physiological limit) and force appliance on the bracket base. Herein study followed the same methodology as previous FEA dental studies (mentioned in references), with only using the correct material-based type failure criteria and the same assumed isotropy, homogeneity/non-homogeneity and linear-elasticity as boundary conditions as did the previous numerical studies. However, herein study by correctly using the above-mentioned obtained correct qualitative and quantitative results (indirectly validated by correlation with maximum hydrostatic pressure and clinical data). The aim of herein study was to verify a set of rules to make the FEA analysis to provide correct results that are in agreement with clinical data (which is a progress in FEA applied in dentistry), since the previous analysis use a mix of failure criteria and boundary condition that provided clinically contradictory and incorrect results.”

  1. Concern of the reviewer:

“5.      The case studies presented in this article are not clearly explained. Also, the numerical analysis used to evaluate the cases of study should be in dynamic behavior considered chewing force and with dentine anisotropic properties.”

Our response:

  • We thank the reviewer for his/her concern and comments. We do hope that our changes are according to the reviewer‘s remarks.

The methodology described herein is the same as in previous FEA studies mentioned in references. Each model is based on radiological examinations, then 3D models are reconstructed and analyzed. The orthodontic loads and movements are similar with previous studies. The only difference is that besides the intact periodontium models, there are models with a simulated gradual horizontal breakdown (since clinically is important to know how bone loss influence stress distribution and quantitative stresses). The orthodontic load is applied on each tooth, and the stress distribution in PDL is analyzed (and correlated with maximum hydrostatic pressure and clinical knowledge).

Revised text: please see the methodology section.

Herein study analyzed the amount of applied orthodontic force in PDL in both intact and reduced periodontium, assessing the biomechanical behavior both qualitatively and quantitatively. It is the first study of this kind that investigated both intact and reduced periodontium. Since all previous FEA studies are with the same methodology as herein, and in order to have the possibility to make correlations, we needed this approach. Moreover, before simulating more complex types of biomechanical behavior, we needed to establish the correct set of rules applicable to dental studies to produce correct results. The reviewer referred to dynamic behavior which is not the subject of herein but will be the subject of further study.

Revised text: pg. 3 lines 142-150:

“Herein aims were a) to assess qualitatively and quantitatively the differences between T (non-homogenous) and VM (homogenous) failure criterion by simulating 0-8 mm periodontal breakdown under five orthodontic movements (extrusion, intrusion, rotation, tipping and translation) and three loads (0.6, 1.2 and 2.4 N) in PDL, b) to verify the use of linear-elasticity, isotropy, homogeneity/non-homogeneity as boundary conditions for more than 1 N loads for PDL, c) to correlate the quantitative results with MHP and available clinical knowledge, evaluating ischemic and resorptive risks for more than 1 N orthodontic loads in PDL.”

Regarding the anisotropic properties, all dental tissues are naturally anisotropic. However, since it is recognized that dental FEA studies cannot accurately reproduce the clinical conditions, all previous numerical available studies assumed the isotropy for dental tissues. Moreover, there are various contradictory reports regarding the differences between isotropy vs. anisotropy (as mentioned in the article). From the biomechanical point of view, it is acknowledged that up to 1 N of load due to extremely small deformations and displacements isotropy, linear-elasticity and homogeneity are correct for dental tissues. Herein study, aimed to investigate if higher orthodontic forces (up to 2.4 N) provide the same qualitative results (stress distribution in PDL) as under 1 N (0.6 N in our case) and how the quantitative results change, all correlated with maximum hydrostatic pressure and clinical data.

Revised text: pg.2 lines 92-102

“Anatomically and biomechanically dental tissues are non-homogenous, anisotropic and do not show linear elasticity [23]. However, all dental studies (PDL included) neglected these issues assuming linear-elasticity, homogeneity, and isotropy (as boundary conditions assumptions), obeying Hook’s law, due to easier mathematical equations and lack of awareness. Nevertheless, from the mechanical point of view (common engineering knowledge), up to 1 N of load due to small deformations and displacements all tissues assume linear-elasticity and isotropy [13, 21-24]. However, for more than 1 N there is no available data about the linear elasticity and isotropy assumptions. Regarding the non-homogeneity/homogeneity issue of dental tissues, Tresca criterion is suited for ductile non-homogenous, while VM for ductile homogenous, as correctly reported earlier analysis [1, 21-24]. There are reports of a quantitative difference between these two criteria, with T 15-30% higher than VM. T is more conservative than VM, since it defines a smaller region of elastic behavior in the principal stress state space, being better suited for correct analyses of extremely small complex structures. There are no other FEA studies covering the above issues except our earlier research [1, 21-24].”

                  Pg. 3 lines 134-141

“Only by correctly identifying and employing boundary conditions and proper failure criterion, FEA dental study can become reliable and correct, as in engineering field [21, 22]. In previous comparative studies our team proved that dental tissues have ductile like resemblance and that only VM and T criteria supply accurate results [1, 21-24]. Only one other older FEA analysis approached the issue of proper failure criteria but for brittle like root canal filling, correlating the failure criterion with the analyzed material type (which is mandatory in engineering field) [2]. No other FEA studies approached these extremely critical issues for FEA accuracy.”

  1. Concern of the reviewer:

“6.      The boundary conditions specify how the solution of the problem behaves at the boundaries of the domain. This is crucial to correctly define the problem being addressed and ensure that the solution is meaningful and applicable to the given context. For this reason, proper boundary conditions can help ensure the numerical stability of methods used to solve partial differential equations or other mathematical problems. Inadequate conditions can lead to numerically unstable or divergent solutions. Also, Boundary conditions should reflect the actual physical conditions of the problem being modeled. For example, in teeth tissue, the boundary conditions could represent fixed gum tissue edges and specified chewing forces. Would you mind explaining in depth the numerical stability of all the cases of study that have been developed in this research, the physical consistency, the mesh stability, and the Solution accuracy? Could you summarize it in a table?.”

Our response:

  • We thank the reviewer for his/her concern and comments. We do hope that our changes are according to the reviewer‘s remarks.

As explained by the reviewer, the boundary conditions help in solving the equations. However, in dentistry the knowledge that makes the method so accurate in engineering field lacks. Moreover, the FEA was developed for the engineering field and had been adopted by dentistry in the las decade or so, but without properly understanding (the brittle or ductile behavior, or what means hydrostatic pressure and why cannot be used for PDL). This is why there are so many FEA studies employing brittle like failure criteria/or hydrostatic pressure for ductile like materials. There is no clear set of rules for making a dental numerical analysis to provide accurate/correct results that are in line with physiological constants (e.g. maximum hydrostatic pressure) and with clinical knowledge. Herein analysis, by performing a FEA analysis using the dental common knowledge regarding the FEA and employing the methodology available in previous dental studies verified if correct results can be obtained. Thus, we needed to start with the basics: the correct failure criteria based on the material type, and isotropy, linear-elasticity and homogeneity/non-homogeneity and small loads of around 1 N with extremely small displacements and deformations and verify their biomechanical correctness of PDL. The results were in line with both MHP and clinical knowledge for all models. Fixed boundary conditions/encastered were imposed at the bottom of the models (i.e. the translations of the FE nodes were restricted, thus also the rotations are zero). In our understanding these conditions are representative and relevant for the studied issues

Revised text: pg. 5 lines: 205-216

“The boundary conditions assumptions were isotropy, homogeneity, linear elasticity (as in all above-mentioned FEA studies), with perfectly bonded interfaces and base of the models encastered (Figure 2). Fixed boundary conditions (i.e., encastered) were imposed at the bottom of the models (i.e. the translations of the FE nodes were restricted, thus also the rotations are zero). In our understanding these conditions are representative and relevant for the studied issues. The failure criterion used were non-homogeneous ductile materials Tresca and homogenous ductile materials Von Mises. The load manager conditions selection: step procedure-static, general; load type-pressure; load status-created in step, distribution-uniform; magnitude-depending on the surface area; amplitude-ramp. Abaqus boundary condition manager selection: step procedure-static, general; boundary condition type-symmetry/antisymmetric/encastre; boundary condition status-created in step.”

  1. Concern of the reviewer:

“7.      Really, you have done the FEM mesh with all stl gaps. I assume that the size of finite elements are too small. For that reason, the time solutions depend on the computer used. Could You please mention the computer that was used to resolve those cases of study.”

Our response:

  • We thank the reviewer for his/her concern and comments. We do hope that our changes are according to the reviewer‘s remarks.

As mentioned to previous above comment, the FEA in dentistry has no clear set of rules. The models used for dental FEA analysis are mostly idealized models of tooth and periodontium, that have almost nothing in common with the anatomical reality (neither as anatomical form or number of elements and nodes). Moreover, there are already dental studies reporting differences between models with small and large number of elements/nodes (less anatomically accurate). Thus, to be able to obtain correct FEA results, we needed the analyzed models to be anatomically accurate. The human patients were X-ray examined, and based on CBCT DICOM slice the 3D model reconstructions were performed. However, the reconstruction process was not performed automatically since the software was not able to automatically recognize and identify the various grey shades of different anatomical components. Thus, all was performed manually by a single doctor. All the internal control checks were performed in order to pass the quality check. Each software has an internal algorithm that allows a model to be reconstructed and further processed if there are no errors. In our models we found only surface irregularities and no element errors. All surface irregularities were placed in areas not interfering with stress distribution, while all other surfaces were quasi-continuous. For e.g., one of the models with the highest number of elements and nodes (6 million elements) had a total of 264 element warnings.

Revised text: pg. 4 lines 184-197

“The mesh models had 5.06-6.05 million C3D4 tetrahedral elements, 0.97-1.07 million nodes, and a global element size of 0.08-0.116 mm (high anatomical accuracy, when compared with above-mentioned FEA studies).

Manual reconstruction displayed a limited number of surface irregularities in all models, located in non-essential areas (i.e., quasi-continuous stress areas) (Figure 2). All internal quality control checking algorithms were successfully passed, with no element errors, and only a limited number of element warnings (e.g., the maximum number of elements warnings was 264 [0.0043%] for a total number of 6.05 million elements). The topographical mesh distribution for elements warnings was 201 (0.0039%) of 5117355 for bone, 63 (0.00677%) of 930023 for tooth-bracket-PDL, 39 (0.00586%) of 665501 for tooth-bracket, 26 (0.00459185%) of 566221 for dentine, and 17 (0.0141469%) of 120168 for enamel-bracket elements. The internal control and safety check of both software (i.e., image reconstruction and finite elements analysis) do not allow the next phase of the process if errors or too many anomalies are present, an aspect that was not seen in our study.”

Regarding the time solution comment, due to the large number of elements and nodes and the small size of elements, the processing time for each model was around 4 hours, on our high-performance DELL computer (Intel Core i7-7700HQ 2.80GHz CPU, 32 GB DDR4 RAM Memory). Although the numerical domain is large (around 6 million FE), the analysis time is reasonable due to the following considerations: linear elastic analysis (no change of structural stiffness), small amount of forces per each load increment thus the stress concentrations are avoided, stable mesh with a negligible number of poor shaped FE, perfect bond between the biological parts.

Revised text: pg.5 lines 216-220

“Although the numerical domain is large (around 6 million FE), the analysis time is reasonable due to the following considerations: linear elastic analysis (no change of structural stiffness), small amount of forces per each load increment thus the stress concentrations are avoided, stable mesh with a negligible number of poor shaped FE, perfect bond between the biological parts.”

  1. Concern of the reviewer:

“8. How was it used in this research? If any biomechanical considerations have not been mentioned. Why?.”

Our response:

  • We thank the reviewer for his/her concern and comments. We do hope that our changes are according to the reviewer‘s remarks.

We think that all biomechanical considerations relevant for herein were mentioned in.

  1. Concern of the reviewer:

“9. The conclusion needs to summarize the significant findings of methods used with future proposals..”

Our response:

  • We thank the reviewer for his/her concern and comments. We do hope that our changes are according to the reviewer‘s remarks.

Revised text: pg.17 lines: 594-610

“1. Both Tresca and Von Mises failure criteria displayed similar qualitative results, while the quantitative ones were 15% higher for T. Since T criteria was designed to describe non-homogenous materials behavior, is better suited for quantitative results than VM, and should be preferably used for FEA simulations.

  1. For all three loads, the qualitative stress display was similar in both failure criteria and for all bone loss levels, seeming that deformations and displacements are constant, and manifested in the same areas independently of the load’s amount, the only difference being their intensity (stress doubled for 1.2 N and quadrupled for 2.4 N).
  2. The boundary conditions assumptions (homogeneity, linear elasticity, isotropy, and perfectly bonded interfaces) seem to be correct up to 2.4 N of applied load, and thus correctly used in FEA analysis of dental tissues.
  3. Clinically speaking, in intact periodontium forces up 2.4 N could be relatively safely applied (mild risks), while in reduced periodontium after 4 mm of loss more than 1 N should be carefully considered (high ischemic and resorptive risks).
  4. Clinically speaking, 4 mm of bone loss seems to be a reference point in orthodontic treatment, since from this moment ischemic and resorptive risks significantly increase, and the benefits must be balanced with risks.”

Regarding the future proposals, more simulations are needed, by improving both the modeling (i.e., internal micro-architecture) technique and boundary conditions assumptions (linear vs. non-linear and anisotropy) in order to enhance the accuracy and transform the FEA in dentistry in a reliable everyday study method.

Revised text: pg. 17 lines 585-592

“To verify the trueness of the above-mentioned boundary conditions and failure criteria more numerical simulations are needed improving both the modeling (i.e., internal micro-architecture) technique and boundary conditions assumptions (linear vs. non-linear and anisotropy) in order to enhance the accuracy and transform the FEA in dentistry in a reliable everyday study method. The accuracy of numerical simulations can be easily improved in dental field by following the requirements set out in the engineering field and by introducing the new AI help (since the complexity of non-linear equations).”

Reviewer 4 Report

Comments and Suggestions for Authors

It was evaluated the article titled “The Importance of Boundary Conditions and Failure Criterion in Finite Elements Analysis Accuracy - A Comparative Assessment of PDL Biomechanical Behavior”.
The goal of this study was “assessing differences between Tresca (T-non-homogenous) and Von Mises (VM-homogenous) criterion by simulating 0-8 mm periodontal breakdown under five orthodontic movements (extrusion, intrusion, rotation, tipping, and translation) and three loads (0.6, 1.2 and 2.4 N). Additionally, addressed the issues of proper boundary conditions selection for more than 1 N loads, and correlated the results with the maximum hydrostatic pressure (MHP) and available knowledge, evaluating ischemic and resorptive risks for more than 1 N orthodontic loads."

The topic studied is very interesting and deserve attention.

“Here FEA run 486 numerical simulations, analyzing 81 second lower premolars models, from nine patients (mean age 29.81 ± 1.45 years, 4 males, 5 females, oral informed consent). Our sample size was nine, more than the above-mentioned FEA studies that used a sample size of one.” I considered 9 a low number too.
Therefore, this phrase above is not scientifically correct. Could the authors provide a scientific response to affirm that 9 is the sufficient number for significant results?

Moreover, I considered the mean age low… better, the LPD is this mean age is different of older patients. Could you provide a justification for it?

Why this region was selected (“The region of interest was the mandibular lateral region with the two molars and premolars”)? Why not to include all teeth (only 9 patients)?

RESULTS
Lines 212-213 “qualitative and quantitative results for PDL for both criteria. No visible influence of age, sex or gender was seen." This fact is because your sample is extremely small. Provide a sample size calculation
(you have numbers already published to achieve it)

The authors provide many numbers and I appreciate it. Therefore, where is the statistical analysis to show any significance?

Discussion is long. Shorten it, please.

Conclusion: revisit e revise the conclusion… how could you affirm as you did? Explain, please.

Author Response

Department of Cariology, Endodontics and Oral Pathology

School of Dental Medicine

University of Medicine and Pharmacy

Ms. Betty Pei

Section Managing Editor

Applied Sciences-Applied Dentistry and Oral Sciences      

Special Issue - Oral Diseases and Clinical Dentistry      

                                                                                                                                 March 5th, 2024

Dear Ms. Betty Pei,

Thank you very much for your letter dated February 27th, 2024, with the comments of the reviewers. We have now carefully considered the comments of the reviewers and amended the paper accordingly. All changes are highlighted in red throughout the manuscript and included also below.

Reply to Reviewer #4:

We agree and we thank the reviewer for his/her time and comments. Appropriate changes in the manuscript have by now been made. Please see below and in the manuscript.

Concern of the reviewer:

” Comments and Suggestions for Authors

It was evaluated the article titled “The Importance of Boundary Conditions and Failure Criterion in Finite Elements Analysis Accuracy - A Comparative Assessment of PDL Biomechanical Behavior”.
The goal of this study was “assessing differences between Tresca (T-non-homogenous) and Von Mises (VM-homogenous) criterion by simulating 0-8 mm periodontal breakdown under five orthodontic movements (extrusion, intrusion, rotation, tipping, and translation) and three loads (0.6, 1.2 and 2.4 N). Additionally, addressed the issues of proper boundary conditions selection for more than 1 N loads, and correlated the results with the maximum hydrostatic pressure (MHP) and available knowledge, evaluating ischemic and resorptive risks for more than 1 N orthodontic loads."

The topic studied is very interesting and deserve attention.

“Here FEA run 486 numerical simulations, analyzing 81 second lower premolars models, from nine patients (mean age 29.81 ± 1.45 years, 4 males, 5 females, oral informed consent). Our sample size was nine, more than the above-mentioned FEA studies that used a sample size of one.” I considered 9 a low number too.
Therefore, this phrase above is not scientifically correct. Could the authors provide a scientific response to affirm that 9 is the sufficient number for significant results?

Moreover, I considered the mean age low… better, the LPD is this mean age is different of older patients. Could you provide a justification for it?

Why this region was selected (“The region of interest was the mandibular lateral region with the two molars and premolars”)? Why not to include all teeth (only 9 patients)?

RESULTS
Lines 212-213 “qualitative and quantitative results for PDL for both criteria. No visible influence of age, sex or gender was seen." This fact is because your sample is extremely small. Provide a sample size calculation
(you have numbers already published to achieve it)

The authors provide many numbers and I appreciate it. Therefore, where is the statistical analysis to show any significance?

Discussion is long. Shorten it, please.

Conclusion: revisit e revise the conclusion… how could you affirm as you did? Explain, please..”

Point-by-point response to the reviewer’s comments:

  1. Concern of the reviewer:

““Here FEA run 486 numerical simulations, analyzing 81 second lower premolars models, from nine patients (mean age 29.81 ± 1.45 years, 4 males, 5 females, oral informed consent). Our sample size was nine, more than the above-mentioned FEA studies that used a sample size of one.” I considered 9 a low number too.

Therefore, this phrase above is not scientifically correct. Could the authors provide a scientific response to affirm that 9 is the sufficient number for significant results?

Moreover, I considered the mean age low… better, the LPD is this mean age is different of older patients. Could you provide a justification for it?”

Our response:

  • We thank the reviewer for his/her concern and comments. We do hope that our changes are according to the reviewer‘s remarks.

In clinical studies a sample size must be high, however, in FEA studies the sample size is always small since numerical studies allow multiple simulations and changes in boundary conditions, loads, 3D models, etc. All previously available dentistry FEA analyses used a sample size of one (usually a lower anatomical accuracy quality 3D model, as mentioned in article), conducting only few simulations before drawing any conclusions (usually contradictory from one study to another, disagreeing the clinical data, and exceeding the maximum hydrostatic pressure). In our simulation, the sample size of nine (nine times more than in the other studies), we analyzed 81 models (vs. one or two in the other simulations) and running 486 simulations, before drawing any conclusions. Moreover, the results were in agreement with clinical data and maximum hydrostatic pressure.

Revised text: pg.4 lines 156-159

“Here FEA run 486 numerical simulations, analyzing 81 second lower premolars models, from nine patients (mean age 29.81 ± 1.45 years, 4 males, 5 females, oral informed consent). Our sample size was nine, more than the above-mentioned FEA studies that used a sample size of one (since numerical studies require only a small sample size).”

                      pg.17 lines 582-585

“Moreover, since the input data can be easily changed, the required sample size is extremely small (e.g., sample size of one in above-mentioned FEA studies and of nine here), thus allowing multiple simulations with various results from a small number of patients [1].”

  1. Concern of the reviewer:

“Moreover, I considered the mean age low… better, the LPD is this mean age is different of older patients. Could you provide a justification for it?.”

Our response:

  • We thank the reviewer for his/her concern and comments. We do hope that our changes are according to the reviewer‘s remarks.

The potential patients needing orthodontic treatment and with non-inflamed periodontium for this study were more numerous, however only nine qualified. We needed our patients not to have much bone loss, since we wanted to run a horizontal periodontal breakdown simulation, and we needed intact periodontium models. Older patients usually have a more advanced bone loss, with tooth loss and/or prosthetic works, which is difficult to reconstruct for obtaining intact periodontium models. Moreover, we were focused on having a model with complete teeth. The anatomical micro-architecture does not differ from younger people to older ones, while the other considerations were considered to be not relevant for our biomechanical behavior analysis of PDL.

  1. Concern of the reviewer:

“Why this region was selected (“The region of interest was the mandibular lateral region with the two molars and premolars”)? Why not to include all teeth (only 9 patients)?.”

Our response:

  • We thank the reviewer for his/her concern and comments. We do hope that our changes are according to the reviewer‘s remarks.

The region was selected since there are many FEA studies with molars and incisors, and only a few with premolars. Regarding the number of teeth included in our model, in all other available FEA studies, only one tooth was selected, and we needed to be able to correlate the results. Moreover, more teeth mean huge computer power, making such complex models to be difficult to simulate since one model of 6 million elements takes around two hours of FEA simulation on very good specifications computers Another reason was that in order to have CBCT with small element size, the ROI is extremely small barely covering 4 teeth. A larger CBCT with an entire arch will provide much higher element size, which will reduce the anatomical accuracy of the models. However, by employing more modes (81) and simulations (486) we were able to obtain correct results in agreement with clinical data and maximum hydrostatic pressure.

  1. Concern of the reviewer:

“RESULTS
Lines 212-213 “qualitative and quantitative results for PDL for both criteria. No visible influence of age, sex or gender was seen." This fact is because your sample is extremely small. Provide a sample size calculation
(you have numbers already published to achieve it).”

Our response:

  • We thank the reviewer for his/her concern and comments. We do hope that our changes are according to the reviewer‘s remarks.

Regarding the sample size. As we above mentioned, the FEA studies (all available studies in current literature flow) employ a sample size of one since the numerical simulations do not involve more than a few analyzed models (due to possibilities to change the models and conditions thus obtaining different results). We used a sample size of nine, thus nine times more than the average dental FEA study. Regarding the sample size calculation, it is not working for numerical studies, since they are so different from the clinical ones.

Revised text: pg.4 lines 156-159

“Here FEA run 486 numerical simulations, analyzing 81 second lower premolars models, from nine patients (mean age 29.81 ± 1.45 years, 4 males, 5 females, oral informed consent). Our sample size was nine, more than the above-mentioned FEA studies that used a sample size of one (since numerical studies require only a small sample size).”

                      pg.17 lines 582-585

“Moreover, since the input data can be easily changed, the required sample size is extremely small (e.g., sample size of one in above-mentioned FEA studies and of nine here), thus allowing multiple simulations with various results from a small number of patients [1].”

                      pg.6 lines 241-244

“Herein FEA analysis totaled 486 simulations displaying qualitative and quantitative results for PDL for both criteria.

The qualitative results displayed similar stress distribution for both criterions. There were no differences in stress display among the three loads.”

  1. Concern of the reviewer:

“The authors provide many numbers and I appreciate it. Therefore, where is the statistical analysis to show any significance?”

Our response:

  • We thank the reviewer for his/her concern and comments. We do hope that our changes are according to the reviewer‘s remarks.

Herein is a study in which we registered the quantitative values for nine models with intact periodontium, that each was then subjected to a horizontal periodontal breakdown from 0 to 8 mm of loss, for five most common orthodontic movements. The quantitative values provided in the tables are the average, since all models displayed quantitative values close to each other. This type of display does not require statistical analyses. Moreover, statistical analysis is not used in numerical studies (none of the FEA studies published in the current literature flow use statistical analysis). The average quantitative values were used for correlations with the physiological maximum hydrostatic pressure in order to see ischemic and resorptive risks, and to verify if herein set of rules for FEA in dentistry provide data in agreement with clinical one. Both above mentioned were accomplished.

  1. Concern of the reviewer:

“Discussion is long. Shorten it, please”

Our response:

  • We thank the reviewer for his/her concern and comments. We do hope that our changes are according to the reviewer‘s remarks.

We agree with the reviewer that the discussion is a bit long, however, in order to be able to make the necessary correlations and comparations with other FEA studies, we needed to take this approach. Moreover, since our study is the first of this type and one of the aims was to be able to verify a set of rules for a dental FEA analysis that provide accurate results, we needed to touch on the most important issues regarding the numerical problems. The shortage of this section would not be in the best interest of the reader, since will lose clarity and coherence, and will not be able to provide a complete picture of the problem. Additionally, other reviewers suggested adding further comments regarding the FEA limits and other biomechanical issues in order to enhance clarity.

  1. Concern of the reviewer:

“Conclusion: revisit e revise the conclusion… how could you affirm as you did? Explain, please..”

Our response:

  • We thank the reviewer for his/her concern and comments. We do hope that our changes are according to the reviewer‘s remarks.

The conclusions are in correlations with the aims of the study. We used FEA analysis employing two failure criteria (T-non-homogenous and VM-homogenous) and isotropy, linear-elasticity and homogeneity/non-homogeneity as boundary conditions, for loads from 0.6 N to 2.4 N, simulating periodontal breakdown from 0-8 mm in PDL and five orthodontic movements. One of the aims was to verify a set of rules to be used in FEA dental studies for providing correct results, since the main problem of numerical studies is that usually the results contradict the clinical knowledge. We accomplished this aim.

Another aim was to analyze the stress distribution in PDL (intact and reduced) both qualitatively and quantitatively in order to be able to make correlations with physiological constants (MHP) and clinical knowledge and based on above to evaluate the ischemic and resorptive risks if higher force are used in reduced periodontium. We accomplished that aim also, and obtained results that are in agreement with clinical knowledge and practice.

Revised text: pg.17 lines 594-610

“1.       Both Tresca and Von Mises failure criteria displayed similar qualitative results, while the quantitative ones were 15% higher for T. Since T criteria was designed to describe non-homogenous materials behavior, is better suited for quantitative results than VM, and should be preferably used for FEA simulations.

  1. For all three loads, the qualitative stress display was similar in both failure criteria and for all bone loss levels, seeming that deformations and displacements are constant, and manifested in the same areas independently of the load’s amount, the only difference being their intensity (stress doubled for 1.2 N and quadrupled for 2.4 N).
  2. The boundary conditions assumptions (homogeneity, linear elasticity, isotropy, and perfectly bonded interfaces) seem to be correct up to 2.4 N of applied load, and thus correctly used in FEA analysis of dental tissues.
  3. Clinically speaking, in intact periodontium forces up 2.4 N could be relatively safely applied (mild risks), while in reduced periodontium after 4 mm of loss more than 1 N should be carefully considered (high ischemic and resorptive risks).
  4. Clinically speaking, 4 mm of bone loss seems to be a reference point in orthodontic treatment, since from this moment ischemic and resorptive risks significantly increase, and the benefits must be balanced with risks.”

Round 2

Reviewer 2 Report

Comments and Suggestions for Authors

Thank you for all the corrections you made. I still have one objection, however.

Revised text: pg.4 lines:160-165

 “The including criteria were complete mandibular dental arch in region of interest, with no malposition and intact teeth in the analyzed region, no advanced bone loss, non-inflamed periodontium, orthodontic treatment indication, proper oral hygiene. The exclusion criteria were related to the incomplete mandibular arch, malposition, non-intact teeth, in the region of interest.”

What amazes me is the fact of carrying out a complex scientific proof of the entire work and not understanding such basic mathematical concepts as sets and subsets that should be included in each other. After all, the exclusion criterion is such a set that MUST be included in the set that was derived from the inclusion criterion. For example: if we included teeth without damage to the crown (intact teeth), it is in vain to look for cases of non-intact teeth in the group formed after applying the inclusion criteria, because all that are in our group are intact teeth, so this criterion has no chance of being met. The exclusion criteria should provide such a chance.

I am not the author of the paper, but my impression is that if during the evaluation of the included teeth it was found that the anatomy of the root of the tooth deviated from the standards (it has a different shape or definitely different length for example, with a hooked bend of the root apex) then such a tooth would be removed from the analysis. "A tooth root anatomy with a different shape or length" - could be an exclusion criterion, not all that the authors present. 

Author Response

Department of Cariology, Endodontics and Oral Pathology

School of Dental Medicine

University of Medicine and Pharmacy

Mr. Paritat Thaitalay

Section Managing Editor

Applied Sciences-Applied Dentistry and Oral Sciences      

Special Issue - Oral Diseases and Clinical Dentistry      

                                                                                                                               March 13th, 2024

Dear Mr. Paritat Thaitalay,

Thank you very much for your letter dated March 11th, 2024, with the comments of the reviewers. We have now carefully considered the comments of the reviewers and amended the paper accordingly. All changes are highlighted in red throughout the manuscript and included also below.

Reply to Reviewer #2:

We agree and we thank the reviewer for his/her time and comments. Appropriate changes in the manuscript have by now been made. Please see below and in the manuscript.

Concern of the reviewer:

Thank you for all the corrections you made. I still have one objection, however.

Revised text: pg.4 lines:160-165

 “The including criteria were complete mandibular dental arch in region of interest, with no malposition and intact teeth in the analyzed region, no advanced bone loss, non-inflamed periodontium, orthodontic treatment indication, proper oral hygiene. The exclusion criteria were related to the incomplete mandibular arch, malposition, non-intact teeth, in the region of interest.”

What amazes me is the fact of carrying out a complex scientific proof of the entire work and not understanding such basic mathematical concepts as sets and subsets that should be included in each other. After all, the exclusion criterion is such a set that MUST be included in the set that was derived from the inclusion criterion. For example: if we included teeth without damage to the crown (intact teeth), it is in vain to look for cases of non-intact teeth in the group formed after applying the inclusion criteria, because all that are in our group are intact teeth, so this criterion has no chance of being met. The exclusion criteria should provide such a chance.

I am not the author of the paper, but my impression is that if during the evaluation of the included teeth it was found that the anatomy of the root of the tooth deviated from the standards (it has a different shape or definitely different length for example, with a hooked bend of the root apex) then such a tooth would be removed from the analysis. "A tooth root anatomy with a different shape or length" - could be an exclusion criterion, not all that the authors present."

Point-by-point response to the reviewer’s comments:

  1. Concern of the reviewer:

“Line 1 – please – choose the type of your manuscript.”

Our response:

  • We thank the reviewer for his/her concern and comments. We do hope that our changes are according to the reviewer ‘s remarks. We agree with the reviewer that the inclusion criteria help identify the study population while the exclusion criteria are the factors that make the recruited population ineligible for the study. We made the text clearer.

In our study the including criteria were complete mandibular dental arch in region of interest, with no malposition and intact teeth (no endodontic treatment, no filling or crown) in the analysed region, no advanced bone loss, non-inflamed periodontium, orthodontic treatment indication, and proper oral hygiene. The non-suitable patients were related to the incomplete mandibular arch, malposition, non-intact teeth, in the region of interest. From the patients that were included in the study less common root geometry (e.g., non-fused double rooted, angulated root, extreme curvature of root, etc.), anormal shape of the crown, deciduous teeth, abnormal root (e.g., external root resorption) or bone shape (various types of bone defects), abnormal pulp chamber (internal resorption), bone loss more than 2-3 mm, any signs of inflamed periodontium or bad oral hygiene after the acceptance in the study were also considered among the exclusion criteria. All teeth that were included had root morphology and dimensions that were considered normal and common in the clinical practice (since our aim was to obtain correct information regarding the clinical biomechanical behaviour and not the idealised anatomy).

Revised text: pg.4 lines 186-197

“The including criteria were complete mandibular dental arch in region of interest, with no malposition and intact teeth (no endodontic treatment, no filling or crown) in the analyzed region, no advanced bone loss, non-inflamed periodontium, orthodontic treatment indication, proper oral hygiene. The non-suitable patients were related to incomplete mandibular arch, malposition, non-intact teeth, in the region of interest. From the patients that were included in the study less common root geometry (e.g., non-fused double rooted, angulated root, root extreme curvature etc.), anormal shape of the crown, deciduous teeth, abnormal root surface defects (e.g., external root resorption) or bone shape (various types of bone defects radiological visible), abnormal pulp chamber (internal resorption, radiologically identified), bone loss of more than 2-3 mm, any signs of inflamed periodontium or bad oral hygiene after the acceptance in the study were also considered among the exclusion criteria.”

Pg:5 lines 215-216

“The nine second premolar models included seven single-rooted and three with fused double rooted.”

Reviewer 4 Report

Comments and Suggestions for Authors

I re-evaluated the article titled “The Importance of Boundary Conditions and Failure Criterion in Finite Elements Analysis Accuracy - A Comparative Assessment of Periodontal Ligament Biomechanical Behavior”.

Where is the sample size calculation requested?
Why pre-molar region? Include more sites

There was no scientific justification for many of the questions raised previously.
For pratically all questions previously raised, the authors had a response or excuse.
- e.g., site selected: “The region was selected since there are many FEA studies with molars and incisors, and only a few with premolars.” - If you developed a study using molars or incisors, the comparison could be easier.

“Regarding the number of teeth included in our model, in all other available FEA studies, only one tooth was selected, and we needed to be able to correlate the results.” It is not a scientific justification.

- my previous comment: “qualitative and quantitative results for PDL for both criteria. No visible influence of age, sex or gender was seen." This fact is because your sample is extremely small."
Authors’ response: “Regarding the sample size. As we above mentioned, the FEA studies (all available studies in current literature flow) employ a sample size of one since the numerical simulations do not involve more than a few analyzed models (due to possibilities to change the models and conditions thus obtaining different results). We used a sample size of nine, thus nine times more than the average dental FEA study. Regarding the sample size calculation, it is not working for numerical studies, since they are so different from the clinical ones.”
Please, revise your concept about the affirmation above. It was a excuse for the question raised. Present a scientific base to justify n = 9, please.

Author Response

Department of Cariology, Endodontics and Oral Pathology

School of Dental Medicine

University of Medicine and Pharmacy

Mr. Paritat Thaitalay

Section Managing Editor

Applied Sciences-Applied Dentistry and Oral Sciences      

Special Issue - Oral Diseases and Clinical Dentistry      

                                                                                                                                 March 13th, 2024

Dear Mr. Paritat Thaitalay,

Thank you very much for your letter dated March 11th, 2024, with the comments of the reviewers. We have now carefully considered the comments of the reviewers and amended the paper accordingly. All changes are highlighted in red throughout the manuscript and included also below.

Reply to Reviewer #4:

We agree and we thank the reviewer for his/her time and comments. Appropriate changes in the manuscript have by now been made. Please see below and in the manuscript.

Concern of the reviewer:

”I re-evaluated the article titled “The Importance of Boundary Conditions and Failure Criterion in Finite Elements Analysis Accuracy - A Comparative Assessment of Periodontal Ligament Biomechanical Behavior”.

Where is the sample size calculation requested?

Why pre-molar region? Include more sites

There was no scientific justification for many of the questions raised previously.

For pratically all questions previously raised, the authors had a response or excuse.

- e.g., site selected: “The region was selected since there are many FEA studies with molars and incisors, and only a few with premolars.” - If you developed a study using molars or incisors, the comparison could be easier.

“Regarding the number of teeth included in our model, in all other available FEA studies, only one tooth was selected, and we needed to be able to correlate the results.” It is not a scientific justification.

- my previous comment: “qualitative and quantitative results for PDL for both criteria. No visible influence of age, sex or gender was seen." This fact is because your sample is extremely small."

Authors’ response: “Regarding the sample size. As we above mentioned, the FEA studies (all available studies in current literature flow) employ a sample size of one since the numerical simulations do not involve more than a few analyzed models (due to possibilities to change the models and conditions thus obtaining different results). We used a sample size of nine, thus nine times more than the average dental FEA study. Regarding the sample size calculation, it is not working for numerical studies, since they are so different from the clinical ones.”

Please, revise your concept about the affirmation above. It was a excuse for the question raised. Present a scientific base to justify n = 9, please.?”

Our response:

We thank the reviewer for his/her concern and comments; however, we totally disagree with reviewer’s remarks regarding our previous responses of avoiding the answer and not providing scientific justification but only excuses. In our previous response we correctly responded to each issue raised.

  1. Regarding the first issue: “the sample size calculation” -We explained that finite elements analyses are numerical descriptive studies that fundamentally differ from the clinical studies, and thus do not obey the same rules (the absence of sample size calculations being one of these), this is why all the current FEA dental studies available in the current research flow do not have sample size calculation (as explaining in both manuscript and our previous response). Moreover, as we mentioned in the previous response, almost all current research flow FEA studies (some of them also mentioned in references) have a sample size of one (one patient and one model, and no more than a few simulations), and yet they were able to formulate results and conclusions. Yet, we agree that more patients and models mean better results and conclusion, and that is why we increased the sample size to nine (nine patients, 81 models, 486 simulations). Our analysis is a descriptive and observational one, in which forces are applied and stress display is assessed both qualitatively and quantitatively and then correlated with clinical data. Regarding the number nine, only nine patients qualified based on inclusion and exclusion criteria. Regarding the sentence “No visible influence of age, sex or gender was seen”, it is true since nothing was observed.

Revised text: pg. 18. lines 616-628

“In numerical studies by changing the parameter, a complete new set of results can be obtained [1, 21-24]. Thus, since the input data can be easily changed, the required sample size is extremely small (e.g., sample size of one in above-mentioned FEA studies [3-19] vs. nine herein), thus allowing multiple simulations with various results from a small number of patients [1, 21-24]. FEA are descriptive studies fundamentally different from clinical ones (different set of rules and requirements, with sample size being one of them [1, 21-24]), clearly visible is numerical dental studies methodology from the current research flow (i.e., sample size of one - one patient/model and few simulations [3-19] vs. nine patients, 81 models and 486 simulations herein). Herein analysis has a sample size nine times higher than most FEA studies [3-19] since we agree that more patients and models imply more simulations and data that produce better results and conclusions [1, 21-24].”

Pg. 4 lines: 186-197:

“The including criteria were complete mandibular dental arch in region of interest, with no malposition and intact teeth (no endodontic treatment, no filling or crown) in the analysed region, no advanced bone loss, intact teeth, non-inflamed periodontium, orthodontic treatment indication, proper oral hygiene. The non-suitable patients were related to incomplete mandibular arch, malposition, non-intact teeth, in the region of interest. From the patients that were included in the study less common root geometry (e.g., non-fused double rooted, angulated root, root extreme curvature etc.), abnormal shape of the crown, deciduous teeth, abnormal root surface defects (e.g., external root resorption) or bone shape (various types of bone defects radiological visible), abnormal pulp chamber (internal resorption, radiologically identified), bone loss of more than 2-3 mm, any signs of inflamed periodontium or bad oral hygiene after the acceptance in the study were also considered among the exclusion criteria”

  1. Regarding the second issue: “Why pre-molar region” – We explained that we are interested to study the premolar region since the lack of studies regarding this region (most of the studies were focused on molar and incisors), the clinical importance from both orthodontic and periodontal point of view and to fulfil the aim of any scientific research to enhance the scientifical knowledge and help the clinician in everyday practice (please see clinical implications).

Revised text: pg. 3 lines 121-134

 “Most of FEA current research flow [3-19] studied intact periodontium models of upper and lower first molars and upper central incisors subjected to a limited amount of orthodontic loads and one or two movements, providing data limited to this anatomical region. Moreover, there are no FEA studies to analysis all five most common orthodontic movements and their tissular comparative impact. Few numerical studies have investigated the premolar area and all for intact periodontium [13, 14], although various levels of bone loss are relatively common in everyday clinical practice. Moreover, in orthodontic field, to know how bone loss changes the biomechanical stress distribution and how to reduce/keep the amount of applied loads in order to avoid ischemia and further tissular loss, is mandatory [1, 21-24]. Thus arises the natural scientific need to study the tissular biomechanics of this region during the periodontal breakdown process. Moreover, since the need for a clear image of biomechanical behavioural changes determined by the bone loss in orthodontic treatment, a gradual horizontal periodontal breakdown process needs to be studied.”

  1. 1 lines 13-34

“Featured Application: For a clinician knowing the amounts of load that can be safely applied during the periodontal breakdown helps in improving the predictability of the orthodontic treatment and avoiding the ischemic and resorptive risks. Thus, knowing that intact periodontium could bear up to 2.4 N without major ischemic or resorptive risks is of extreme importance. The 4 mm breakdown reference point, after which the applied loads should be lower than 1 N, supplies valuable data for both orthodontics and periodontology. The stress distribution areas displayed for each movement and bone loss level create a general complete image of PDL biomechanical behavior.

     For a researcher providing a way to gain the much-needed results accuracy for dental studies comparable with those provided by the engineering field is valuable, since FEA is the only available method that allows individual study of each dental tissues’ component and the current numerical studies produced debatable and sometimes contradictory results. Thus, by employing the ductile resemblance material type failure criteria (T and VM), and linear-elasticity, isotropy, and homogeneity/non-homogeneity as boundary conditions assumptions in the study of PDL, herein study obtained results which are in agreement with clinical knowledge. Moreover, the above-mentioned boundary conditions are correct up to an applied load of 2.4 N (up to 1 N being acknowledged to be mechanically correct).”

  1. Regarding the second issue: “Include more sites” – We explained that it was not our objective to include the entire mandibular arch due to three reasons: First is related to the accuracy of the anatomic model (our models are based on DICOM slices of 0.0075 mm) since it is not radiologically possible to acquire such extended ROI for and obtain at least a voxel size of 0.075 mm, thus, we would lose the accuracy of the models and the simulations will not meet the FEA requirements. Moreover, the molar and incisor region have been already studied (as above-mentioned). The Second is related to the fact that a large mandibular arch model imply not only advanced simulations of the biomechanical movements that actually are not possible in FEA programs (since do not have yet developed algorithms), but also will demand an enormous compute power (hardware) that we do not possess (our DELL computers have Intel Core i7-7700HQ 2.80GHz CPU, 32 GB DDR4 RAM Memory, 32 GB dedicated video memory and takes between 4-6 hours of processing for a single model). The third is related to the fact the molar and incisors region have been already studied.

Revised text: pg.3 lines 135-145

“Most FEA studies [3-19] usually investigated a model of only one tooth since the difficulty in performing numerical simulations on models with multiple teeth [1, 21-24]. Field et al. [14] simulated a mandibular arch model with three teeth subjected to orthodontic movement reporting however unnatural and clinically incorrect qualitative stress displays, due to high element size and reduced number of elements-nodes of the analysed model. To have a correct anatomical model, means to be based on CBCT images of in vivo tissues, recorded of at least 0.075 mm voxel size, but that implies restraining the recorded field. Moreover, a model with a single tooth anatomically accurate (i.e., extremely small global element size and high number of elements and nodes) needs extremely high compute power [1, 21-24]. Another issue is related to the fact that there are no algorithms to test complex tissular biomechanics.”
